# Learning Structure-Dynamics-aware Representations for Efficient and Robust 3D Pose Estimation

## Abstract

Recent works in 2D-to-3D pose uplifting for monocular 3D Human Pose Estimation (HPE) have shown significant progress. However, two key challenges persist in real-world applications: vulnerability to joint noise and high computational costs. These issues arise from the dense joint-frame connections and iterative correlations typically employed by mainstream GNN-based and Transformer-based methods. To address these challenges, we propose a novel approach that leverages human physical structure and long-range dynamics to learn spatial part- and temporal frameset-based representations. This method is inherently robust to missing or erroneous joints while also reducing model parameters. Specifically, in the Spatial Encoding stage, coarse-grained body parts are used to construct structural correlations with a fully adaptive graph topology. This spatial correlation representation is integrated with muti-granularity pose attributes to generate a comprehensive pose representation for each frame. In Temporal Encoding and Decoding stages, Skipped Self-Attention is performed in framesets to establish long-term temporal dependencies from multiple perspectives of movement. On this basis, a compact Graph and Skipped Transformer (G-SFormer) is proposed, which realises efficient and robust 3D HEP in both experimental and practical scenarios. Extensive experiments on Human3.6M, MPI-INF-3DHP and Human-Eva benchmarks demonstrate that G-SFormer series models can compete and outperform the state-of-the-arts but takes only a fraction of parameters and around 1% computational cost. It also exhibits outstanding robustness to inaccurately detected 2D poses. The source code will be available at *sites.google.com/view/g-sformer*.

## 1 Introduction

3D Human Pose Estimation (HPE) is a fundamental task which aims to reconstruct 3D body joint locations from images or videos. Monocular 3D HPE is more friendly for downstream applications such as action recognition (Shi et al., 2019b; Yan et al., 2018; Shi et al., 2019a), human-computer interaction (Sinha et al., 2010; Lazar et al., 2017; Pavlovic et al., 1997), motion and trajectory prediction (Martinez et al., 2017; Wang et al., 2021; Rudenko et al., 2020) for the convenience in data acquisition.

Benefits from rapid development in 2D pose detectors (Chen et al., 2018; Sun et al., 2019), 2D-to-3D pose lifting methods have drawn extensive attentions for its high spatial precision and light data volume of 2D skeletons. Despite the superior performance, 2D-to-3D lifting method is inherently an ill-pose problem for depth ambiguity and self-occlusion (Cheng et al., 2019; 2020; Li et al., 2022b). To alleviate this issue, recent works focused on aggregating temporal motion information in videos to aid pose reconstruction. Transformer has become a prevalent approach for its long-range dependency modeling capacity. Prior methods typically deploy self-attention to establish joint-wise correlations, as well as frame-wise correlations for each joint individually or for the encoded pose representation (Zhang et al., 2022; Zheng et al., 2021). This is computationally expensive especially when dealing with lengthy sequences (81, 243 or even more), rendering it impractical for deployment on resource-limited mobile devices and consumer hardware. Given the information redundancy in adjacent frames, improving the efficiency of Transformers becomes an essential issue. Additionally, inaccuracies in detected 2D poses, including missing and erroneous joints also present

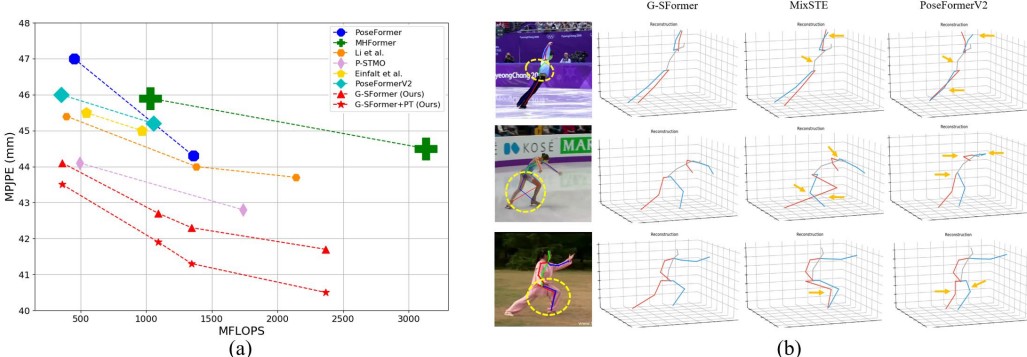

Figure 1: (a) MPJPE (mm) vs. MFLOPs of G-SFormer and competitors on Human3.6M, where marker size indicates model size. (b) Robustness comparison with MixSTE (Zhang et al., 2022) and PoseFormerV2 (Zhao et al., 2023) regarding errors in detected 2D joints (marked in yellow circles).

challenges to 3D HPE. Factors such as self-occlusion, fast-motion, as well as unconstrained environment and image quality significantly increase the error rate of detected 2D joints, making model robustness an important property in practical applications. Some efforts have been made on these two challenges, Li et al. (2022a) and Shan et al. (2022) introduce strided convolutional layers to FFN in transformer to selectively aggregate useful information; Einfalt et al. (2023) perform temporal upsampling on encoded pose sequence to realise efficient pose uplifting from sparse tokens; Zhao et al. (2023) obtain low-frequency pose components through filtering out high-frequency noise and integrate them with sampled temporal features to generate a robust pose representation. Li et al. (2024) propose token pruning cluster module to select a few representative tokens to reduce temporal redundancy. However, it is worth noting that reducing pose tokens in temporal domain will lead to performance degradation since partial information is inevitably lost. Furthermore, none of them cut to the optimization of the biggest computational overhead – the Self Attention calculation which is quadratic to the number of tokens.

In this paper, we propose a Frameset-based Skipped Transformer architecture for temporal feature representation and aggregation. The input pose sequence is sampled into framesets to implement Skipped Self-Attention (SSA) in parallel, establishing global-range alignments across multiple temporal perspectives. As a result, the model acquires a composite understanding of the entire motion process, integrating various aspects of movements. This brings two main benefits: first, it significantly improves model efficiency by establishing correlations among distinct frame tokens; second, it captures long-range dynamics and supplement each frame with enhanced contextual information. For the spatial modeling of pose within each frame, we maintain the global approach by employing coarse-grained body parts to construct a compact Part-based Adaptive GNN. Unlike present methods which compute joint-wise connections (Zhang et al., 2022; Tang et al., 2023; Yu et al., 2023; Peng et al., 2024), our approach builds spatial correlations among body parts to better represent the coordination of human body and the interaction between body parts during movement. For instance, the arms and torso are closely related during "Eating", while the legs are correlated for "Sitting". The part-based graph structure is fully adaptive, learned through a graph attention mechanism without relying on pre-defined skeletal topology as priors (Soroush Mehraban, 2024; Yu et al., 2023; Peng et al., 2024), thereby enhancing model flexibility and generalization across diverse poses. Meanwhile, fine-grained joint features are integrated to the part-based representation to enrich the spatial cues. Thus, a comprehensive pose representation is obtained, incorporating global spatial correlations and multi-granularity pose attributes. It is also a robust representation with less sensitivity to local joint deviations.

The main modules of Part-based Adaptive GNN and Frameset-based Skipped Transformer construct the Graph and Skipped Transformer (G-SFormer) architecture. Different from existing Graph-Transformer hybrid methods (Zhu et al., 2021; Zhao et al., 2022; Soroush Mehraban, 2024) that embed GCN into Transformer block to assist self-attention in spatial modeling, G-SFormer integrates part-based adaptive GNN and Skipped Transformer to efficiently exploit spatial and temporal information, respectively. The compact and adaptive framework realises high accuracy, efficient and robust 3D pose reconstruction in a global approach. Extensive experiments are conducted on three benchmarks, i.e., Human3.6M, MPI-INF-3DHP, and HumanEva. G-SFormer can compete

and outperform the state-of-the-arts, and obtain performance-enhancement with the prior-knowledge brought by direct pre-training on large-scale motion capture data. More importantly, as shown in Figure 1(a), the multi-scale G-SFormer models exhibit steady and advance performance with significantly less computational cost and parameters, making them highly suitable for practical 3D HPE applications on resource-limited platforms. In summary, our main contributions are listed as follows:

- We propose two novel modules of Part-based Adaptive GNN and Frameset-based Skipped Transformer to learn comprehensive pose representations and multi-perspective dynamic representations, enabling efficient and robust 3D pose estimation in a global approach.
- Effective data completion methods are developed to enhance the spatial structure and temporal motion information of input 2D poses, including Sinusoidal Positional Encoding and Data Rolling strategies, which are parameter-free and easy to implement.
- The formulated compact G-SFormer achieves advance performance across 3 large-to-small benchmarks. Multi-scale G-SFormer models can compete and outperform the state-of-the-arts with only a fraction of parameters and around 1% computational cost, offering an effective and practical approach to realise high-accuracy 3D HPE with small-scale model size and minimal computational cost.

## 2 RELATED WORKS

Traditional 2D-to-3D pose lifting includes CNN-based methods (Pavllo et al., 2019; Chen et al., 2021), GCN-based methods(Zeng et al., 2021; Yu et al., 2023), and Transformer-based methods (Zheng et al., 2021; Tang et al., 2023). Recently, the diffusion framework has also been introduced into 3D HPE, delivering outstanding performance through the aggregation and selection among multiple iterations and hypotheses (Holmquist & Wandt, 2023; Shan et al., 2023; Peng et al., 2024). However, the diffusion process also considerably increase computational overhead and inference time. Therefore, we build upon traditional methods and propose a light-weight architecture to realise efficient and robust 3D pose estimation.

### 2.1 GCN-BASED 3D HPE

Recent lifting-based approaches exploit contextual information from neighboring frames to improve robustness and accuracy. Pavllo et al. (2019) present a fully-convolutional model with dilated temporal convolutions to regress temporal information. Chen et al. (2021) leverage CNN-based bone length and bone direction prediction networks to derive 3D joint locations. Due to natural correlations with human skeleton structures, GCN-based methods are adopted to incorporate spatial priors. Cai et al. (2019) construct a spatial-temporal graph to process and consolidate pose features across scales. Wang et al. (2020) design a U-shaped GCN to capture motion information and leverage motion supervision for 3D sequence reconstruction. Similar to 2s-AGCN (Shi et al., 2019b), methods like GLA-GCN and KTPFormer (Yu et al., 2023; Peng et al., 2024) add a learnable graph to the predefined adjacency matrix based on skeletal structure to improve graph adaptation. However, the predefined graph which is used to maintain performances and stabilize training somewhat restricts model flexibility. Besides, the joint-based graph topology lacks global coordination between body parts and introduces computational redundancy.

### 2.2 TRANSFORMER-BASED 3D HPE

Transformer architecture proposed by Vaswani et al. (2017) has shown promising performance in computer vision (Zhu et al., 2020; Dosovitskiy et al., 2020). With the outstanding ability in capturing long-range dependencies, transformer with powerful self-attention mechanism is introduced to 3D HPE task. Zheng et al. (2021) firstly apply Transformer to model spatial and temporal aspects in 2D-to-3D pose lifting. Considering the characteristics of transformer, pre-training operations such as self-supervised (Shan et al., 2022) and large-scale dataset based (Einfalt et al., 2023; Zhu et al., 2023) are implemented for performance improvement. MixSTE (Zhang et al., 2022) models temporal motion of each joint and stacks spatial and temporal transformer blocks several loops to strength sequence coherence. On this basis, other architectures learning separate single joint motion trajectory have been developed. They either perform spatial and temporal modeling sequentially in

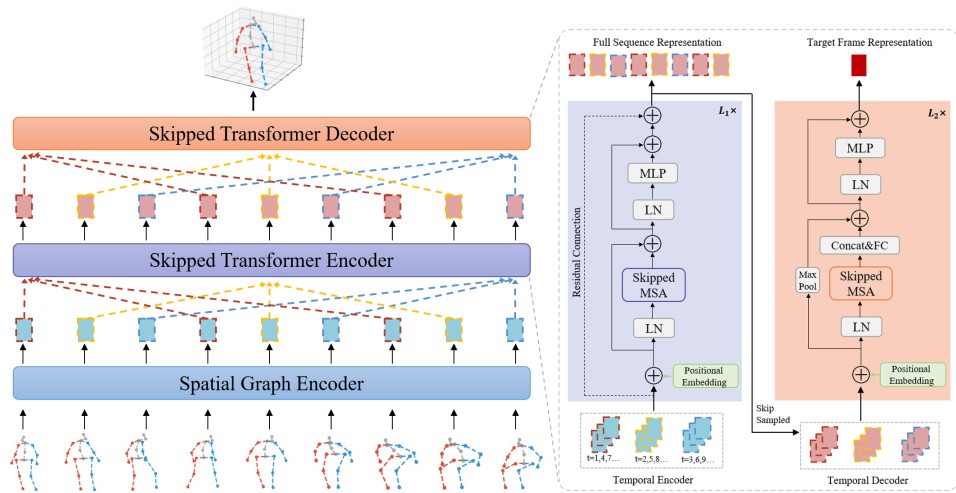

Figure 2: Graph and Skipped Transformer (G-SFormer) consists of three modules: Spatial Graph Encoder for part-based structural learning within each frame, Skipped Transformer Encoder and Decoder for hierarchical extraction and aggregation of temporal features. After being encoded by the Skipped-Transformer, skip-sampled framesets are reordered back to the original sequence and progressively aggregated by Skipped Multi-head Self-Attention (MSA) to get the target pose representation in the temporal decoding stage.

an iterative process (Peng et al., 2024; Soroush Mehraban, 2024; Zhu et al., 2023) or utilize a parallel modeling approach (Tang et al., 2023). However, the repetitive and redundant frame-wise and joint-wise connections make up large and complex networks, introducing considerable computational overhead and increasing model sensitivity to local features.

## 3 METHOD

Following 2D-to-3D pose lifting pipelines (Shan et al., 2022; Tang et al., 2023; Zhao et al., 2023), the proposed G-SFormer regress 3D pose of the center frame from input 2D pose sequence estimated by off-the-shelf 2D pose detectors. As shown in Figure 2, Graph Neural Network (GNN) and Skipped Transformer are engaged as key components for spatio-temporal feature learning.

### 3.1 SPATIAL GRAPH CONSTRUCTION

In the spatial modeling of the 2D pose in each frame, human joints are grouped into 5 parts according to their physical relationships. Without any predefined connections as priors (Hu et al., 2021; Peng et al., 2024; Yu et al., 2023), a flexible graph topology is learned by a totally data-driven approach to better represent the interaction between body parts. As shown in Figure 3 (a), the grouped 2D joint coordinates $l_p$ are passed into a parameter-sharing Parts Encoding Layer composed of Multi-Layer Perceptron (MLP) structure to get part feature $f_p$, which is a $T \times N_p \times C$ tensor, where $T$, $N_p$ and $C$ refer to frame length, number of body parts and channel dimension. A learnable part positional embedding $E_p \in \mathbb{R}^{N_p \times C}$ is added with $f_p$ to integrate the structure information.

$$f_p = MLP(l_p) + E_p \qquad (1)$$

Then, the $N_p \times N_p$ adjacency matrix is obtained by performing attention calculation among part features, where the attention coefficients can be calculated by:

$$e_{i,j} = \mathbf{W} \times |f_{pi} + f_{pj}| \qquad (2)$$

Part features are added to establish alignments with each other. $\mathbf{W}$ is the weight vector of $1 \times 1$ convolution which transforms the dimension of attention map into 1. Attention coefficients are normalized using the $Sigmoid$ function to obtain the inter-part correlation strength.

$$\alpha_{i,j} = Sigmoid(e_{i,j}) = 1/(1 + exp(e_{i,j})) \qquad (3)$$

Figure 3 (b) illustrates the updating process of graph nodes. Part features are aggregated with attention coefficients to obtain the graph feature, which is then concatenated with the original part feature to get the refreshed $f_p^{'}$.

$$f_{pi}^{'} = Concat\left(\sigma(\sum\nolimits_{j=1}^{N_p} \alpha_{i,j} f_{pj}), f_{pi}\right) \quad (4)$$

$\sigma$ is the nonlinear activation function, and $\|$ represents concatenation. In order to preserve the fine-grained spatial information, joint feature encoded by Fully Connected layer is residually added to $f_p^{'}$ to obtain the comprehensive pose representation $F_P \in \mathbb{R}^{T \times D}$.

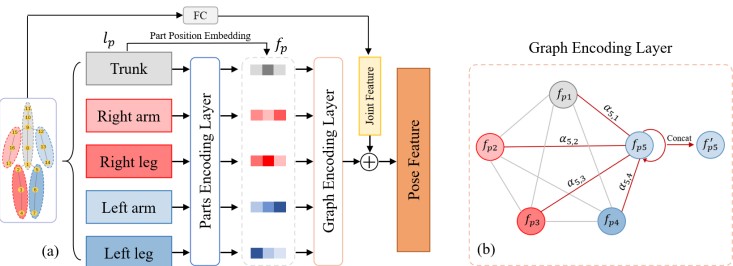

(a)      (b)

Figure 3: (a) Architecture of the Spatial Graph Encoder. (b) Updating process of graph nodes. Using part feature $f_{p5}$ as an example, it is concatenated with the aggregated part features weighted by attention coefficients to get the refreshed $f_{p5}^{'}$.

## 3.2 SKIPPED TRANSFORMER FOR TEMPORAL MODELING

Skipped Transformer-based Encoder and Decoder are deployed in the temporal modeling process. Both modules work in a computationally efficient manner by concurrently establishing long-range dynamics in multiple framesets. Corresponding complexity analysis of them is also presented.

### 3.2.1 SKIPPED TRANSFORMER FOR TEMPORAL ENCODING

The proposed Skipped Self-Attention (SSA) builds global dependencies among distinct frame tokens. Specifically, skipped connection with interval $m$ is performed on temporal dimension to construct long-range attention alignments. The sampling process is conducted $m$ times until all tokens are established associations.

$$Attn(Q_i, K_i, V_i) = Softmax(\frac{Q_i K_i^T}{\sqrt{D}})V_i$$
$$= A_i V_i, i = 1, 2, ...m \quad (5)$$

where $Q_i$, $K_i$ and $V_i$ are obtained by linearly transformation of skipped sampling frameset $Z_i \in \mathbb{R}^{\frac{T}{m} \times D}$ with parameters $W_Q, W_K$ and $W_V$.

$$Q_i = Z_i W_Q, K_i = Z_i W_K, V_i = Z_i W_V \quad (6)$$

The frameset consists of $\frac{T}{m}$ tokens and the attention map $A_i$ is with the shape of $(\frac{T}{m}) \times (\frac{T}{m})$. Therefore, the computational complexity for attention map calculation and token association is significantly reduced to $\frac{1}{m}$ times compared with the Self-Attention in Vanilla Transformer. Thus the compression ratio can be derived as 30%, as skipped factor $m$ is set to 3 in the proposed architecture.:

$$\Omega(SSA) = \frac{2T^2 D}{m} \quad (7)$$

Finally, the encoded frame tokens in each sampling set are reordered back to original sequence to keep temporal dimension unchanged. Similar to Vaswani et al. (2017), the multi-head setting for self-attention and the feed-forward network is also deployed in the Skipped Transformer block. The temporal encoder is composed of a stack of $L_1$ layers for hierarchical temporal feature extraction.

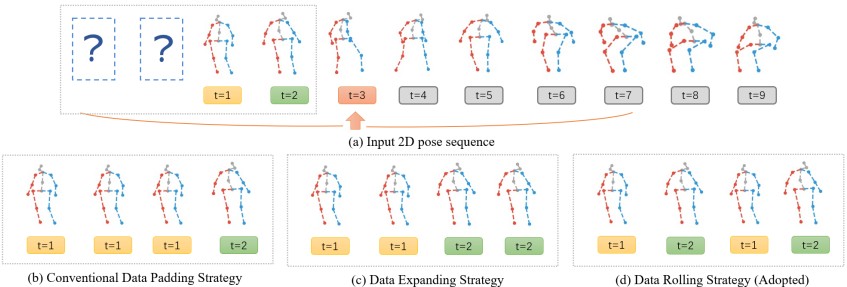

Figure 4: Data completion strategies for 2D pose input. Taking target frame at t=3 as example, where 2 previous frames need to be completed for a full 9-frame input sequence. Unlike conventional methods which copy edge frame at t=1 (b), Data Expanding and Data Rolling strategies are proposed to replicate 2D pose step by step (c), or to capture a clip of the pose sequence for completion (d).

### 3.2.2 SKIPPED TRANSFORMER FOR TEMPORAL DECODING

In the decoding stage, the encoded pose feature sequence is aggregated by layers using Skipped Self-Attention (SSA). Unlike the encoding stage, $m$ skipped sampling framesets are concatenated in the channel dimension, generating a decoded feature sequence with the shape of $\frac{T}{m} \times (m \cdot D)$, which is then linearly transformed to $\frac{T}{m} \times D$. Thus, the temporal dimension is progressively reduced by $\frac{1}{m}$ through each decoding layer until the center frame representation is obtained in the last $L_2$ layer.

Skipped Transformer aggregates temporal features directly in self-attention calculation. It offers a simple yet effective way compared to the widely-adopted Strided Transformer (Einfalt et al., 2023; Shan et al., 2022; Li et al., 2022a). For Skipped Transformer block (SKT), the complexity for key components is:

$$
\begin{aligned}
\Omega(SKT) &= \frac{2T^2D}{m} + m(\frac{T}{m})D^2 + 4(\frac{T}{m})D^2 \\
&= \frac{2T^2D}{m} + (1 + \frac{4}{m})TD^2
\end{aligned}
\tag{8}
$$

Where $\frac{2T^2D}{m}$ is for self-attention as equation 7, $m(\frac{T}{m})D^2$ is for linear transformation, and $4(\frac{T}{m})D^2$ is for feed-forward network (FFN). While for Strided Transformer block (STT) which uses strided convolution layer with the kernel size $k$ and strided factor $s$ to shrink sequence length, the complexity for self-attention and strided convolutional FFN is:

$$
\Omega(STT) = 2T^2D + 2(\frac{k}{s} + 1)TD^2
\tag{9}
$$

Quantitatively, comparing a SKT with $m$=3 and a STT with $k$=3 and $s$=3, the computational cost of the former is less than 60% of the latter under the same setting.

### 3.3 DATA COMPLETION STRATEGIES FOR 2D POSE INPUT

We perform data completion in both spatial and temporal domains. Sinusoidal Positional Encoding is introduced to supplement the relative positional relationships and differences between body joints in 2D pose input. This spatial positional encoding is conducted in an economic parameter-free manner as follows, where J denotes the number of joints:

$$
E_{j,x} = sin(j), E_{j,y} = cos(j), j = 1, 2, ...J
\tag{10}
$$

To reconstruct 3D pose of the center frame, 2D poses from (T-1)/2 previous and subsequent frames are required to form an input sequence with length T. Hence, target frames at the beginning and end of the video suffer from missing 2D poses. The previous method involved replicating frames at $t = 1$ or $t = T$ like edge padding (Tang et al., 2023; Zheng et al., 2021; Shan et al., 2022). However, we argue that such monotonous information has a limited help to 3D pose reconstruction. As shown

in Figure 4, we propose Data Expanding and Data Rolling strategies to enrich dynamic information. A threshold $R$ is set for Data Rolling strategy. Specifically, when the length of 2D poses to be completed exceeds $R$, the data completion operation is executed. Performance comparisons of Data Rolling with various $R$ and Data Expanding are provided in Section 4.4.

### 3.4 LOSS FUNCTION

Regression heads consisting of linear transformation layers are deployed to transform full-sequence representation of Temporal Encoder $Z_{EN} \in \mathbb{R}^{T \times D}$ and target frame representation of Temporal Decoder $Z_{DE} \in \mathbb{R}^{1 \times D}$ into corresponding 3D body joint locations. We use L2 loss to conduct full-sequence supervision of encoder outputs, as well as the target-frame supervision of decoder outputs. The full-to-single supervision strategy (Li et al., 2022a; Zheng et al., 2021) benefits the optimization process and introduces temporal consistency to feature learning. Thus, the architecture is trained in an end-to-end manner with the objective loss function as:

$$\mathcal{L} = \mathcal{L}_t + \lambda \mathcal{L}_f \tag{11}$$

In detail, target frame loss and full sequence loss are as follows. Where $p$ and $p^{gt}$ denote the predicted and ground truth 3D joints. $\lambda$ is the balance factor:

$$\mathcal{L}_t = \frac{1}{J} \sum_{i=1}^{J} \left\| p_i - p_i^{gt} \right\|, \quad \mathcal{L}_f = \frac{1}{T} \frac{1}{J} \sum_{t=1}^{T} \sum_{i=1}^{J} \left\| p_{t,i} - p_{t,i}^{gt} \right\|$$

## 4 EXPERIMENTS

### 4.1 DATASETS

The proposed architecture is evaluated on three 3D HPE benchmarks, i.e., Human3.6M (Ionescu et al., 2013), MPI-INF-3DHP (Mehta et al., 2017), and HumanEva (Sigal et al., 2010). Detailed descriptions and evluation metrics on these datasets are presented in Appendix.

### 4.2 IMPLEMENTATION DETAILS

The smaller, standard and larger G-SFormer are presented as G-SFormer-S, G-SFormer and G-SFormer-L on Human3.6M, with encoder-layer ($L1$), decoder-layer ($L2$) set as (3, 5), (4, 5), (8, 5) for 243 frames input. Residual connection across encoder layers is conducted only for G-SFormer-L. For G-SFormer-S with 81 frames and 27 frames input on MPI-INF-3DHP and Human-Eva, ($L1$, $L2$) is set as (3, 4) and (3, 3), respectively. We adopt the 2D pose detected by CPN (Chen et al., 2018) on Human3.6M following (Zhao et al., 2023; Einfalt et al., 2023) and ground truth data on MPI-INF-3DHP and Human-Eva following (Peng et al., 2024; Zhang et al., 2022). More detailed experimental settings are described in Appendix.

### 4.3 COMPARISON WITH STATE OF THE ARTS

#### 4.3.1 QUANTITATIVE COMPARISON FOR EFFICIENCY

**Results on Human3.6M** We compare the proposed G-SFormer with SOTA approaches on Human3.6M dataset in Table 1. Performance among 15 action categories under MPJPE (Protocol #1) and P-MPJPE (Protocol #2) are reported. G-SFormer-L obtains an average MPJPE of 41.6mm and an average P-MPJPE of 33.5mm. With the additional knowledge gained from pre-training on large-scale motion dataset AMASS (Mahmood et al., 2019; Einfalt et al., 2023), performance improves to 40.5mm and 32.5mm, respectively. The MPJPE rivals the second-best competitor, STCFormer (Tang et al., 2023) but with only 40% parameters and 1.5% computational cost. It also achieves comparable P-MPJPE with the best scores across various actions. By further incorporating the reprojection refinement post-processing (Cai et al., 2019; Einfalt et al., 2023; Shan et al., 2022), G-SFormer-L achieves the MPJPE of 39.9mm, outperforming the current best competitor (Peng et al., 2024) by 0.2mm and obtains the lowest error in 8 out of 15 actions.

Since G-SFormer is proposed to realise efficient 3D HPE, the comprehensive properties of model size, computational cost and performance are the focus of assessment. Table 2 presents properties in

Parameter number, FLOPs, and MPJPE of G-SFormer and competitors. G-SFormer-S trained from scratch achieve 1.9 mm-2.5 mm lower MPJPE, while utilizing similar FLOPs and only 30.4%-35% parameters compared to PoseFormer-V2 (Zhao et al., 2023), which already demonstrates an excellent speed-accuracy trade-off among competitors. G-SFormer-S also matches the performance of MixSTE-81f (Zhang et al., 2022) with only 14.9% parameters and 1.2% computational cost. The cost reduction is even more significant compared to the current-best KTPFormer (Peng et al., 2024) with G-SFormer-L requiring 22.7% of parameters and merely 0.8% of FLOPs of the latter. We confirm that the pretraining process enhances the generalization of small-scale G-SFormer models and brings performance improvement. Furthermore, the above results validate that G-SFormer offers an effective solution for high-accuracy 3D HPE with small-scale model size and minimal computational cost of around 1% compared to the state-of-the-art methods.

Table 1: Quantitative comparisons with SOTA methods on Human3.6M of MPJPE (mm) under Protocol #1 and P-MPJPE (mm) under Protocol #2, using CPN detected 2D poses as input. +PT denotes using pre-training process, (*) refers to the refinement module from Cai et al. (2019). Best: **bold**, second best: underlined.

| MPJPE | | Dir. | Disc. | Eat | Greet | Phone | Photo | Pose | Pur. | Sit | SitD. | Smoke | Wait | WalkD. | Walk | WalkT. | Avg. |
|---|---|---|---|---|---|---|---|---|---|---|---|---|---|---|---|---|---|
| Pavllo et al. Pavllo et al. (2019) (T=243) | CVPR'19 | 45.2 | 46.7 | 43.3 | 45.6 | 48.1 | 55.1 | 44.6 | 44.3 | 57.3 | 65.8 | 47.1 | 44.0 | 49.0 | 32.8 | 33.9 | 46.8 |
| Cai et al. Cai et al. (2019) (T=7)(*) | ICCV'19 | 44.6 | 47.4 | 45.6 | 48.8 | 50.8 | 59.0 | 47.2 | 43.9 | 57.9 | 61.9 | 49.7 | 46.6 | 51.3 | 37.1 | 39.4 | 48.8 |
| Liu et al. Liu et al. (2020) (T=243) | CVPR'20 | 41.8 | 44.8 | 41.1 | 44.9 | 47.4 | 54.1 | 43.4 | 42.2 | 56.2 | 63.6 | 45.3 | 43.5 | 45.3 | 31.3 | 32.2 | 45.1 |
| UGCN Wang et al. (2020) (T=96) | ECCV'20 | 40.2 | 42.5 | 42.6 | 41.1 | 46.7 | 56.7 | 41.4 | 42.3 | 56.2 | 60.4 | 46.3 | 42.2 | 46.2 | 31.7 | 31.0 | 44.5 |
| Chen et al. Chen et al. (2021) (T=243) | TCSVT'21 | 41.4 | 43.5 | 40.1 | 42.9 | 46.6 | 51.9 | 41.7 | 42.3 | 53.9 | 60.2 | 45.4 | 41.7 | 46.0 | 31.5 | 32.7 | 44.1 |
| PoseFormer Zheng et al. (2021) (T=81) | ICCV'21 | 41.5 | 44.8 | 39.8 | 42.5 | 46.5 | 51.6 | 42.1 | 42.0 | 53.3 | 60.7 | 45.5 | 43.3 | 46.1 | 31.8 | 32.2 | 44.3 |
| MHFormer Li et al. (2022b) (T=351) | CVPR'22 | 39.2 | 43.1 | 40.1 | 40.9 | 44.9 | 51.2 | 40.6 | 41.3 | 53.5 | 60.3 | 43.7 | 41.1 | 43.8 | 29.8 | 30.6 | 43.0 |
| Li et al. Li et al. (2022a) (T=351)(*) | TMM'22 | 40.3 | 43.3 | 40.2 | 42.3 | 45.6 | 52.3 | 41.8 | 40.5 | 55.9 | 60.6 | 44.2 | 43.0 | 44.2 | 30.0 | 30.2 | 43.7 |
| P-STMO Shan et al. (2022) (T=243)(*) | ECCV'22 | 38.4 | 42.1 | 39.8 | 40.2 | 45.2 | 48.9 | 40.4 | 38.3 | 53.8 | 57.3 | 43.9 | 41.6 | 42.2 | 29.3 | 29.3 | 42.1 |
| MixSTE Zhang et al. (2022) (T=243) | CVPR'22 | 37.6 | 40.9 | 37.3 | 39.7 | 42.3 | 49.9 | 40.1 | 39.8 | 51.7 | 55.0 | 42.1 | 39.8 | 41.0 | 27.9 | 27.9 | 40.9 |
| PoseFormerV2 Zhao et al. (2023) | CVPR'23 | - | - | - | - | - | - | - | - | - | - | - | - | - | - | - | 45.2 |
| GLA-GCN Yu et al. (2023) (T=243) | ICCV'23 | 41.3 | 44.3 | 40.8 | 41.8 | 45.9 | 54.1 | 42.1 | 41.5 | 57.8 | 62.9 | 45.0 | 42.8 | 45.9 | 29.4 | 29.9 | 44.4 |
| STCFormer Tang et al. (2023) (T=243) | ICCV'23 | 38.4 | 41.2 | 36.8 | 38.0 | 42.7 | 50.5 | 38.7 | 38.2 | 52.5 | 56.8 | 41.8 | 38.4 | 40.2 | 26.2 | 27.7 | 40.5 |
| Einfalt et al. Einfalt et al. (2023) (T=351, +PT)(*) | WACV'23 | 39.6 | 43.8 | 40.2 | 42.4 | 46.5 | 53.9 | 42.3 | 42.5 | 55.7 | 62.3 | 45.1 | 43.0 | 44.7 | 30.1 | 30.8 | 44.2 |
| KTPFormer Peng et al. (2024) (T=243) | CVPR'24 | 37.3 | 39.2 | 35.9 | 37.6 | 42.5 | 48.2 | 38.6 | 39.0 | 51.4 | 55.9 | 41.6 | 39.0 | 40.0 | 27.0 | 27.4 | 40.1 |
| G-SFormer-S (T=243) | Ours | 40.3 | 43.2 | 39.6 | 40.8 | 43.9 | 50.1 | 41.6 | 40.1 | 53.1 | 60.0 | 43.3 | 41.1 | 43.4 | 29.8 | 30.0 | 42.7 |
| G-SFormer-S (T=243, +PT) | Ours | 39.5 | 42.3 | 38.4 | 39.4 | 44.1 | 49.5 | 40.3 | 40.1 | 52.1 | 59.2 | 42.8 | 40.2 | 42.3 | 29.1 | 29.7 | 41.9 |
| G-SFormer (T=243) | Ours | 39.8 | 42.9 | 39.2 | 40.2 | 43.3 | 49.9 | 41.2 | 39.6 | 53.0 | 59.9 | 43.0 | 40.3 | 42.6 | 29.4 | 29.8 | 42.3 |
| G-SFormer (T=243, +PT) | Ours | 37.8 | 41.9 | 37.1 | 39.5 | 43.6 | 47.9 | 40.3 | 38.6 | 53.0 | 58.9 | 42.6 | 39.6 | 41.5 | 28.7 | 29.1 | 41.3 |
| G-SFormer-L (T=243) | Ours | 38.9 | 42.3 | 39.1 | 39.4 | 43.6 | 49.3 | 40.9 | 38.6 | 52.0 | 56.4 | 42.0 | 39.6 | 41.9 | 29.5 | 29.8 | 41.6 |
| G-SFormer-L (T=243, +PT) | Ours | 38.3 | 40.7 | 37.0 | 38.8 | 42.6 | 47.8 | 39.3 | 38.3 | 50.1 | 57.0 | 41.5 | 38.6 | 40.5 | 28.4 | 28.6 | 40.5 |
| G-SFormer-L (T=243, +PT)(*) | Ours | 36.8 | 40.1 | 37.2 | 37.5 | 42.4 | 46.4 | 39.0 | 37.7 | 49.9 | 55.6 | 40.8 | 38.5 | 40.1 | 28.2 | 28.4 | 39.9 |
| **P-MPJPE** | | Dir. | Disc. | Eat | Greet | Phone | Photo | Pose | Pur. | Sit | SitD. | Smoke | Wait | WalkD. | Walk | WalkT. | Avg. |
| PoseFormer Zheng et al. (2021) (T=81) | ICCV'21 | 32.5 | 34.8 | 32.6 | 34.6 | 35.3 | 39.5 | 32.1 | 32.0 | 42.8 | 48.5 | 34.8 | 32.4 | 35.3 | 24.5 | 26.0 | 34.6 |
| Li et al. Li et al. (2022a) (T=351)(*) | TMM'22 | 32.7 | 35.5 | 32.5 | 35.4 | 35.9 | 41.6 | 33.0 | 31.9 | 45.1 | 50.1 | 36.3 | 33.5 | 35.1 | 23.9 | 25.0 | 35.2 |
| P-STMO Shan et al. (2022) (T=243)(*) | ECCV'22 | 31.3 | 35.2 | 32.9 | 33.9 | 35.4 | 39.3 | 32.5 | 31.5 | 44.6 | 48.2 | 36.3 | 32.9 | 34.4 | 23.8 | 23.9 | 34.4 |
| MixSTE Zhang et al. (2022) (T=243) | CVPR'22 | 30.8 | 33.1 | 30.3 | 31.8 | 33.1 | 39.1 | 31.1 | 30.5 | 42.5 | 44.5 | 34.0 | 30.8 | 32.7 | 22.1 | 22.9 | 32.6 |
| PoseFormerV2 Zhao et al. (2023) | CVPR'23 | - | - | - | - | - | - | - | - | - | - | - | - | - | - | - | 35.6 |
| GLA-GCN Yu et al. (2023) (T=243) | ICCV'23 | 32.4 | 35.3 | 32.6 | 34.2 | 35.0 | 42.1 | 32.1 | 31.9 | 45.5 | 49.5 | 36.1 | 32.4 | 35.6 | 23.5 | 24.7 | 34.8 |
| STCFormer Tang et al. (2023) (T=243) | ICCV'23 | 29.3 | 33.0 | 30.7 | 30.6 | 32.7 | 38.2 | 29.7 | 28.8 | 42.2 | 45.0 | 33.3 | 29.4 | 31.5 | 20.9 | 22.3 | 31.8 |
| Einfalt et al. Einfalt et al. (2023) (T=351, +PT)(*) | WACV'23 | 32.7 | 36.1 | 33.4 | 36.0 | 36.1 | 42.0 | 33.3 | 33.1 | 45.4 | 50.7 | 37.0 | 34.1 | 35.9 | 24.4 | 25.4 | 35.7 |
| KTPFormer Peng et al. (2024) (T=243) | CVPR'24 | 30.1 | 32.3 | 29.6 | 30.8 | 32.3 | 37.3 | 30.0 | 30.2 | 41.0 | 45.3 | 33.6 | 29.9 | 31.4 | 21.5 | 22.6 | 31.9 |
| G-SFormer (T=243) | Ours | 31.9 | 34.6 | 32.6 | 33.6 | 33.9 | 39.3 | 32.2 | 31.0 | 42.9 | 47.6 | 35.1 | 32.1 | 34.2 | 24.0 | 24.3 | 33.9 |
| G-SFormer (T=243, +PT) | Ours | 30.7 | 34.0 | 30.8 | 32.9 | 33.7 | 38.6 | 31.7 | 30.1 | 42.3 | 47.3 | 34.8 | 31.7 | 33.3 | 23.4 | 23.9 | 33.3 |
| G-SFormer-L (T=243) | Ours | 31.0 | 34.2 | 32.4 | 33.1 | 34.0 | 38.7 | 32.0 | 30.6 | 42.4 | 45.4 | 34.6 | 31.4 | 33.7 | 24.2 | 24.2 | 33.5 |
| G-SFormer-L (T=243, +PT) | Ours | 30.5 | 33.0 | 30.2 | 32.2 | 33.0 | 37.5 | 31.0 | 30.0 | 41.0 | 45.9 | 33.9 | 30.7 | 32.6 | 23.1 | 23.6 | 32.5 |

Table 2: Quantitative comparisons with SOTA methods on Human3.6M under Parameter number, FLOPs, and MPJPE (mm). (+PT) indicates models with additional pre-training stage. Best: **bold**, second best: underlined.

| Method | | Frames | Params (M) | FLOPs (M) | MPJPE↓ |
|---|---|---|---|---|---|
| PoseFormer Zheng et al. (2021) | ICCV'21 | 27 | 9.59 | 452 | 47.0 |
| PoseFormer Zheng et al. (2021) | ICCV'21 | 81 | 9.60 | 1358 | 44.3 |
| MHFormer Li et al. (2022b) | CVPR'22 | 27 | 18.92 | 1030 | 45.9 |
| MHFormer Li et al. (2022b) | CVPR'22 | 81 | 19.70 | 3132 | 44.5 |
| Li et al. Li et al. (2022a) | TMM'22 | 81 | 4.06 | 392 | 45.4 |
| Li et al. Li et al. (2022a) | TMM'22 | 243 | 4.23 | 1372 | 44.0 |
| Li et al. Li et al. (2022a) | TMM'22 | 351 | 4.34 | 2142 | 43.7 |
| P-STMO-S Shan et al. (2022) +PT | ECCV'22 | 81 | 5.4 | 493 | 44.1 |
| P-STMO Shan et al. (2022) +PT | ECCV'22 | 243 | 6.7 | 1737 | 42.8 |
| Einfalt et al. Einfalt et al. (2023) +PT | WACV'23 | 81 | 10.36 | 543 | 45.5 |
| Einfalt et al. Einfalt et al. (2023) +PT | WACV'23 | 351 | 10.39 | 966 | 45.0 |
| PoseFormerV2 Zhao et al. (2023) | CVPR'23 | 81 | 14.35 | 352 | 46.0 |
| PoseFormerV2 Zhao et al. (2023) | CVPR'23 | 243 | 14.35 | 1055 | 45.2 |
| MixSTE Zhang et al. (2022) | CVPR'22 | 81 | 33.65 | 92692 | 42.7 |
| MixSTE Zhang et al. (2022) | CVPR'22 | 243 | 33.65 | 278076 | 40.9 |
| STCFormer Tang et al. (2023) | ICCV'23 | 81 | 4.75 | 13070 | 42.0 |
| STCFormer-L Tang et al. (2023) | ICCV'23 | 243 | 18.91 | 156392 | 40.5 |
| KTPFormer Peng et al. (2024) | CVPR'24 | 81 | 33.65 | 92706 | 41.8 |
| KTPFormer Peng et al. (2024) | CVPR'24 | 243 | 33.65 | 278119 | 40.1 |
| G-SFormer-S/ +PT | Ours | 81 | 4.37 | 361 | 44.1/43.5 |
| G-SFormer-S/ +PT | Ours | 243 | 5.02 | 1092 | 42.7/41.9 |
| G-SFormer/ +PT | Ours | 243 | 5.54 | 1346 | 42.3/41.3 |
| G-SFormer-L/ +PT | Ours | 243 | 7.65 | 2366 | 41.7/40.5 |

**Results on MPI-INF-3DHP** Table 3 reports the performance on the MPI-INF-3DHP. With 27 frames input, G-SFormer-S trained from scratch outperforms MixSTE (Zhang et al., 2022) by large margins of 3.7%, 12.7% and 27.5mm in PCK, AUC and MPJPE, while occupying only 0.4% computational cost and 11% parameters. Even with the substantial reduction in complexity, G-SFormer/-S

are able to achieve the second and third best results for PCK and MPJPE. The stable performance further confirms the efficacy of our method.

**Results on HumanEva** To further explore the generalization of our method, we conduct experiments on small HumanEva dataset and present results in Table 4. G-SFormer-S is trained from scratch and achieves the best performance with only 13% parameters and 1.2% FLOPs compared to the leading competitor KTPFormer (Peng et al., 2024), demonstrating strong generalization ability from large to small datasets.

Table 3: Quantitative comparisons with SOTA methods on MPI-INF-3DHP dataset. +PT indicates models with additional pre-training stage. Best: **bold**, second best: underlined.

| Method | | PCK↑ | AUC↑ | MPJPE↓ |
|---|---|---|---|---|
| Chen et al. Chen et al. (2021) (T=81) | TCSVT'21 | 87.9 | 54.0 | 78.8 |
| PoseFormer Zheng et al. (2021) (T=9) | ICCV'21 | 88.6 | 56.4 | 77.1 |
| P-STMO Shan et al. (2022) (T=81) | ECCV'22 | 97.9 | 75.8 | 32.2 |
| MixSTE Zhang et al. (2022) (T=27) | CVPR'22 | 94.4 | 66.5 | 54.9 |
| Einfalt et al. Einfalt et al. (2023) (T=81) | WACV'23 | 95.4 | 67.6 | 46.9 |
| Einfalt et al. Einfalt et al. (2023) (T=81, +PT) | WACV'23 | 97.1 | 70.0 | 41.2 |
| GLA-GCN Yu et al. (2023) (T=81) | ICCV'23 | 98.5 | 79.1 | 27.8 |
| STCFormer Tang et al. (2023) (T=81) | ICCV'23 | 98.7 | 83.9 | 23.1 |
| PoseFormerV2 Zhao et al. (2023) (T=81) | CVPR'23 | 97.9 | 78.8 | 27.8 |
| KTPFormer Peng et al. (2024) (T=81) | CVPR'24 | **98.9** | **85.9** | **16.7** |
| G-SFormer-S (T=27) | Ours | 98.1 | 79.2 | 27.4 |
| G-SFormer-S (T=81) | Ours | 98.5 | 80.4 | 25.5 |
| G-SFormer (T=243, +PT) | Ours | 98.7 | 82.0 | 23.1 |

Table 4: Quantitative comparisons with SOTA methods on Human-Eva dataset of MPJPE (mm) under Protocol #1. +PT indicates models finetuned from the Human3.6M pre-training models and (†) is our re-implementation results.

| Protocol #1 | Walk | | | Jog | | | |
|---|---|---|---|---|---|---|---|
| | S1 | S2 | S3 | S1 | S2 | S3 | Avg. |
| PoseFormer Zheng et al. (2021) (T=43) | 16.3 | 11.0 | 47.1 | 25.0 | 15.2 | 15.1 | 21.6 |
| PoseFormer Zheng et al. (2021) (T=43, +PT) | 14.4 | 10.2 | 46.6 | 22.7 | 13.4 | 13.4 | 20.1 |
| MixSTE Zhang et al. (2022) (T=43) | 16.2 | 14.2 | 21.6 | 24.6 | 23.2 | 25.8 | 20.9 |
| MixSTE Zhang et al. (2022) (T=43, +PT) | 12.7 | 10.9 | **17.6** | 22.6 | 15.8 | 17.0 | 16.1 |
| PoseFormer-V2 Zhao et al. (2023) (T=81)(†) | 18.3 | 12.9 | 35.1 | 28.9 | 16.4 | 17.7 | 21.5 |
| KTPFormer Peng et al. (2024) (T=43) | 16.5 | 13.9 | 19.9 | 25.3 | 15.9 | 16.5 | 18.0 |
| KTPFormer Peng et al. (2024) (T=27) | **12.3** | 11.5 | 19.5 | 20.9 | 13.1 | 14.5 | 15.3 |
| G-SFormer-S (T=81) | **12.3** | **9.0** | 25.1 | **20.3** | **11.0** | **12.1** | **15.0** |

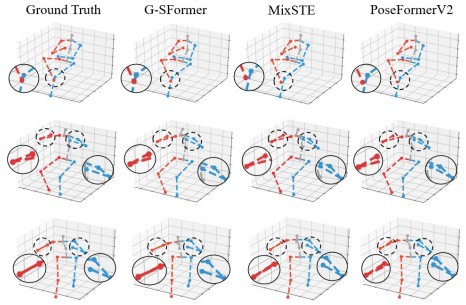

Figure 5: Visualized qualitative comparison with MixSTE (Zhang et al., 2022) and Pose-FormerV2 (Zhao et al., 2023) on Human3.6M dataset.

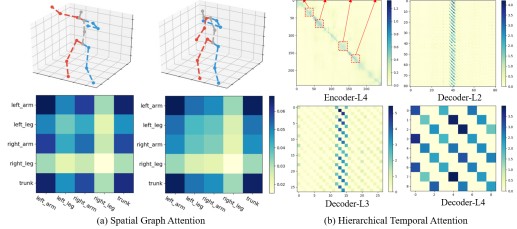

Figure 6: Visualized attention maps of "Greeting" action in (a) Spatial Graph Encoder and (b) Skipped Transformer based Temporal Encoder/Decoder. Attention maps corresponding to multiple heads of Skipped Transformer are summed to produce the holistic temporal correlation distribution.

### 4.3.2 QUALITATIVE COMPARISON FOR ROBUSTNESS

**Visualized Comparison.** In this part, we present the visualized results of G-SFormer and representative competitors (Zhang et al., 2022; Zhao et al., 2023) in and out of the datasets. Figure 5 shows the comparisons in actions of Smoking, Sitting and Photoing from Human3.6M. G-SFormer realise more accurate 3D pose reconstruction across all challenging examples with complex postures.

Videos in-the-wild are even more challenging for 3D HPE task due to complex and changeable movements. Typical hard cases are presented in Figure 1(b), including detection errors such as joint position deviation, left-right switch, confusion caused by self-occlusion and miss/coincidence detection. G-SFormer estimates more refined and structurally reasonable 3D poses than the competitors, demonstrating superiority through part-based alignments that are inherently less sensitivity to local joint deviations, as well as enhanced temporal contextual correlations from multiple aspects of movement. In contrast, MixSTE (Zhang et al., 2022) exhibits higher sensitivity to noise, attributed to its excessive focus on local information while lacking a comprehensive understanding of global motion process within the pose sequence. Although PoseFormerV2 (Zhao et al., 2023) extracts global-view representation by low-frequency DCT coefficients, this compression manner discards features in temporal domain and inevitably causes a loss of accuracy. In contrast, G-SFormer exhibits stable performance in terms of both accuracy and robustness against missing or erroneous joints, with additional qualitative comparisons provided in the Appendix.

**Attention Visualization.** Figure 6 presents spatial and temporal attention distributions. Greeting action of S11 subject on Human3.6M is applied for visualization. It can be seen from (a) that attention

concentrates on left arm, right arm and trunk which are main parts for hugging motion. In the temporal domain (b), global attention is built at full-sequence scale from the attention map of Encoder-L4 layer, and the strength increases during the periods when typical hugging action is performed. As the hierarchical decoding stage progresses, the scope of attentional alighments gradually expand to a global scale as tokens in deeper decoder layers have stronger representation capabilities. *Stronger attention weights distributed sparsely in both local and global temporal intervals, which is significantly different from the dense and clustered attention maps of conventional transformer-based methods (Zheng et al., 2021; Shan et al., 2022).* Specific comparisons are presented in Appendix.

## 4.4 ABLATION STUDY

To verify the effectiveness of main proposals, extensive ablation studies are conducted on Human3.6M. The presented analysis is based on G-SFormer-S with 243 frames input.

Table 5: MPJPE (mm) of G-SFormer-S trained and tested w/ w/o Sinusoidal Positional Encoding (SPE) and different Data Rolling (DR) threshold $R$.

| No. | SPE | DR | MPJPE |
|---|---|---|---|
| 1 | ✗ | R=30 | 42.85 |
| 2 | ✔ | R=10 | 42.87 |
| 3 | ✔ | R=30 | **42.70** |
| 4 | ✔ | R=50 | 42.78 |
| 5 | ✔ | R=90 | 42.85 |
| 6 | ✔ | ✗ | 42.93 |
| 7 | ✗ | ✗ | 43.12 |

Table 6: Ablation studies of the impact of different components and skipped factor $m$ of G-SFormer-S. Experiments are conducted on Human3.6M dataset of MPJPE (mm).

| No. | | m | SSA | Parts Enco | Spatial Attn | FLOPs (M) | MPJPE |
|---|---|---|---|---|---|---|---|
| 1 | Spatial-MLP | 3 | ✔ | ✔ | ✗ | 1100 | 43.6 |
| 2 | Joint-wise GCN | 3 | ✔ | ✗ | ✔ | 1387 | 44.4 |
| 3 | VT-Strided Conv | 1 | ✗ | ✔ | ✔ | 1219 | 43.7 |
| 4 | VT-Conv | 1 | ✗ | ✔ | ✔ | 2111 | 44.0 |
| 5 | G-SFormer-S-m3 | 3 | ✔ | ✔ | ✔ | 1092 | **42.7** |
| 6 | G-SFormer-S-m5 | 5 | ✔ | ✔ | ✔ | 1038 | 43.0 |
| 7 | G-SFormer-S-m7 | 7 | ✔ | ✔ | ✔ | 1016 | 43.8 |
| 8 | G-SFormer-S-m9 | 9 | ✔ | ✔ | ✔ | 997 | 43.9 |

**Data Completion Methods** Table 5 shows the impacts of Sinusoidal Positional Encoding (SPE) and Data Rolling (DR). G-SFormer-S models equipped with SPE and DR show MPJPE error reduction of 0.15 - 0.23mm. On the other hand, the impact of Data Expanding is not obvious, which is less than 0.1mm. To account for datasets of different scales, R is typically set to a ratio of 10% - 20% relative to various input lengths.

**Impact of Components** Table 6 lists the impact of different components and various skipped factor $m$ to overall performance of G-SFormer-S. In row 1 and 2, MLP layers and Joint-wise GCN are used to replace Part-based Adaptive GNN, leading to error increase of up to 1.7 mm. To verify the effectiveness of Skipped Transformer, Vanilla Transformer-based (VT) models are presented in row 2-3, incorporated Convolutional layer in (Zheng et al., 2021; Zhao et al., 2023) and Strided Convolutional layer in (Cai et al., 2019; Shan et al., 2022) for temporal aggregation, respectively. Performance drop 1.0-1.3 mm can be observed with *up to nearly twice computational cost*. It is important to note that G-SFormer shows the highest performance without relying on redundant spatial and temporal connections in Joint-wise GCN and VT series structures, further demonstrating its structural advantages. To validate the effect of factor $m$ in Skipped Transformer, G-SFormer-S with $m = 1, 3, 5, 7, 9$ is evaluated in row 4-8 ($m = 1$ equals to VT-Conv). We draw the conclusion that $m$ should be restrained within an appropriate range, as a proper $m$ strengthens global dependencies and reduces computational complexity. However, excessively high value of $m$ will degrade the temporal coherence of pose sequence. Overall, the hyperparameter $m$ enables G-SFormer with an adaptive architecture for different cost and accuracy requirements.

## 5 CONCLUSION

In this paper, we present a simple yet effective Graph and Skipped Transformer (G-SFormer) architecture for lifting-based 3D HPE task. G-SFormer consists of two main modules for spatial and temporal modeling, respectively. Specifically, a Part-based Adaptive GNN constructs a fully adaptive topology to represent interactions of body parts, and integrates multi-granularity pose attributes to learn a comprehensive pose representation. A Frameset-based Skipped Transformer establish long-range contextual dependencies for identical frames and captures global dynamics by integrating multiple motion variations within framesets. This spatio-temporal global modeling approach significantly reduces computational complexity and alleviates sensitivity to local joints, enhancing robustness against inaccuracies in detected 2D poses. Experiments demonstrate that G-SFormer realises high-accuracy, efficient, and robust 3D HPE for both experimental and wild videos.

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
