# A APPENDIX

## A.1 DATASETS

**Human3.6M** is the most popular indoor 3D HPE dataset which consists of 15 daily activities performed by 11 human subjects. Following the settings of previous works (Li et al., 2022a; Zheng et al., 2021; Zhang et al., 2022), poses of Human3.6M is represented as 17 joint skeletons, subjects of (S1, S5, S6, S7, S8) are used for training and (S9, S11) are used for testing. Evaluation metrics of Mean Per Joint Position Error (MPJPE) and Procrustes analysis MPJPE (P-MPJPE), also known as Protocol #1 and Protocol #2, are presented. MPJPE measures the average Euclidean distance between the estimated 3D joint locations and the ground truth. In P-MPJPE, rigid transformation comprising scale, rotation and translation is applied on estimated 3D pose to align it with the ground truth.

**MPI-INF-3DHP** is also a challenging dataset that contains 3D poses under indoor and outdoor environments. Same as prior SOTA methods (Zheng et al., 2021; Zhang et al., 2022; Peng et al., 2024), metrics of percentage of correct keypoints (PCK) within the 150mm range, area under the curve (AUC) and MPJPE are reported.

**HumanEva** is a small dataset but challenging for model generalization ability. Following the settings of (Zheng et al., 2021; Zhang et al., 2022; Peng et al., 2024), the Walking and Jogging actions of the subjects (S1, S2, S3) are evaluated using MPJPE.

## A.2 IMPLEMENTATION DETAILS

The proposed architecture is implemented in Pytorch with two GeForce RTX 3090 GPUs for training and testing, with a batch size set to 260. The Adam optimizer (Kingma & Ba, 2014) is adopted, with an initial learning rate 1e-3 and a shrink factor of 0.95 per epoch for Human3.6M dataset, while for MPI-INF-3DHP and HumanEva, the shrink factor is set to 0.97. The channel dimension $D$ of the temporal feature sequence is 256, and the balance factor $\lambda$ in the loss function is set to 1.

In the pre-training stage, AMASS (Mahmood et al., 2019) is employed as the meta-dataset. The detailed data processing and transformation methods follow those described in Einfalt et al. (2023), consisting of two key stages. First, the SMPL mesh in AMASS motion data is reduced into J=17 joints with combined 3D joint locations. Second, based on the camera parameter settings from Human3.6M dataset, 3D joint locations are projected into 2D space to generate corresponding 2D pose sequences. In this way, the 2D poses are projected without errors, but can still convey 2D-to-3D pose generation prior knowledge to models with abundant motion data.

## A.3 QUALITATIVE COMPARISON

To further verify the generalizability and robustness of G-SFormer, we provide additional qualitative results across a variety of poses from in-the-wild videos, and make comparison with representative methods in Figure 1. It is worth mentioning that poses in wild videos differ significantly from those in the Human3.6M training set used by G-SFormer. Moreover, factors such as self-occlusion, fast motion, complex and varied movements, as well as detection errors in 2D joints, present considerable challenges for 3D pose estimation. G-SFormer demonstrates superiority in both accuracy and robustness compared to competitive methods that prioritize either high accuracy through dense spatio-temporal connections or robust performance using low-frequency pose representations. Furthermore, given its lightweight model size and low computational cost, G-SFormer holds significant practical value for 3D HPE tasks in complex real-world scenarios.

## A.4 ATTENTION COMPARISON

In this section, we compare the visualized attention maps of Skipped Self-Attention in G-SFormer with the conventional Self-Attention in a typical Transformer-based architecture P-STMO (Shan et al., 2022). While P-STMO also employs an encoder-decoder framework like G-SFormer, it integrates strided convolutional layers into the Vanilla Transformer block for token aggregation during the decoding stage. As shown in Figure 2, intuitive comparisons are facilitated with aligned attention maps in corresponding Encoder/Decoder layers. Based on the multi-perspective modeling of motion

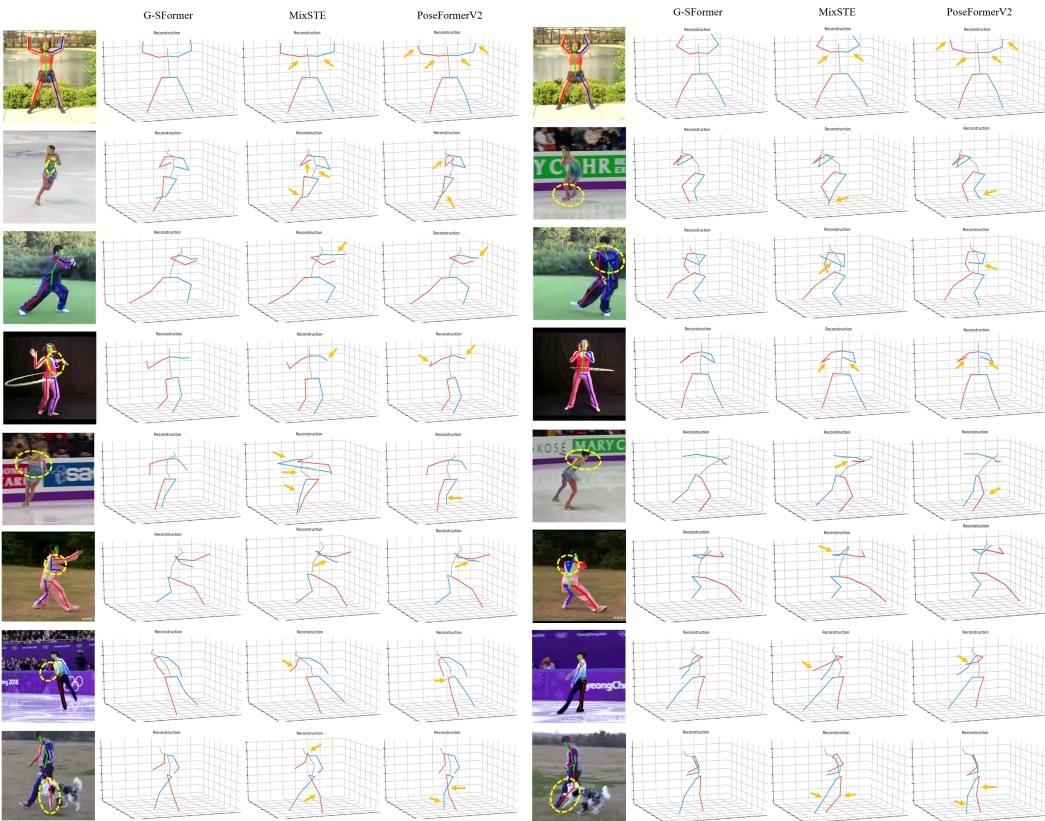

Figure 1: Qualitative Comparison with MixSTE (Zhang et al., 2022) and PoseFormerV2 (Zhao et al., 2023) under challenging in-the-wild videos. The erroneously detected 2D joints are marked in yellow circles and the inaccurately constructed 3D joint locations are marked with arrows. In diverse scenarios, G-SFormer consistently generates more refined 3D estimation results and exhibits stronger robustness to inaccuracies in detected 2D poses.

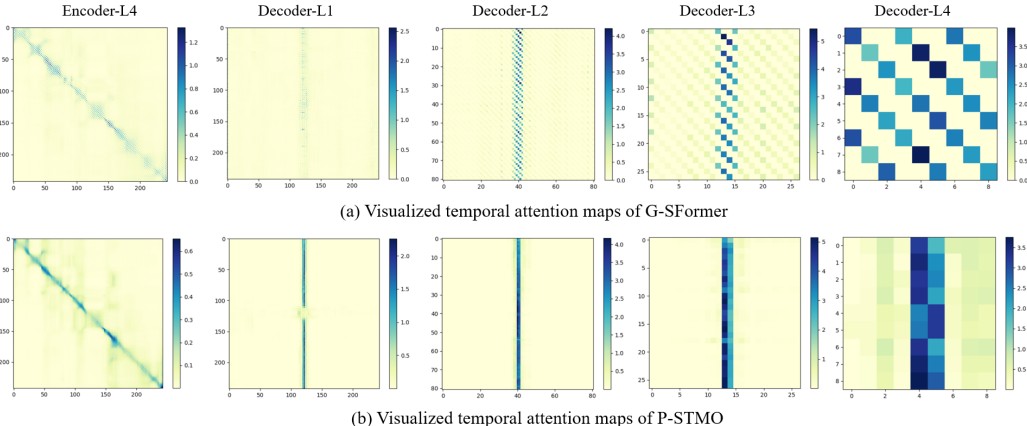

Figure 2: Visualized temporal attention comparison of (a) the proposed G-SFormer and (b) Transformer-based P-STMO. Attention maps are aligned according to the number of temporal tokens in different Encoder/Decoder Layers.

process, the proposed Skipped Self-Attention exhibits sparse attention patterns distributed across a wider temporal range compared to the dense and concentrated self-attention in Vanilla Transformer. Consequently, long-range contextual dependencies are established rather than local connections with

the central frames. The contrast is even more obvious in the deeper layer Decoder-L4 of G-SFormer, where global-range alignments are constructed among all representative tokens. Thus, a comprehensive exploitation of temporal information is achieved with reduced computational cost. The above analysis also indicates that the proposed Skipped Self-Attention demonstrates extensibility in Transformer-based sequential modeling tasks across various fields, such as action recognition and motion prediction.

## A.5 DETAILED QUANTITATIVE COMPARISON FOR EFFICIENCY

We have analyzed the inherent hardware occupation and computational overhead required by G-SFormer and competitors for conducting inference, taking into account total parameter count and FLOPs as discussed in the main manuscript. Based on this, we further incorporate FLOPs/frame to assess the computational cost of generating single-frame 3D pose.

As introduced in the main manuscript, G-SFormer has the Skipped Transformer Encoder and Decoder for temporal feature extraction and aggregation, respectively. Equipped with two independent regression heads, G-SFormer has two outputs: the Encoder predicts the 3D pose sequence corresponding to the entire 2D input sequence, while the Decoder constructs the target 3D pose corresponding to the middle frame of 2D input sequence. This design enables G-SFormer to operate effectively in both *seq2seq* and *seq2frame* workflows.

To further evaluate the capabilities in the *seq2seq* workflow, we supplement the results of the G-SFormer-Encoder. As shown in Table 1, G-SFormer-Encoder models show a significant Parameter reduction of 43.1-60.4%, and a total FLOPs cost reduction of 10.7-18.8% compared to the integral G-SFormer. Notably, the per-frame pose generation cost of G-SFormer-Encoder is far behind of all the existing approaches, ranging from just **4.5/8.7 MFLOPs/frame**. Compared with the best competitor KTPFormer (Peng et al., 2024), it only takes **12.9% of the parameters** and **merely 0.76% of the computational cost**. Despite the minimal computational cost and compact model size, it achieves an MPJPE of 41.6mm, outperforming the low-cost variants of large-scale *seq2seq* methods.

Table 1: Quantitative comparisons with SOTA methods on Human3.6M under Parameter number, FLOPs, FLOPs/frame, and MPJPE (mm). (+PT) indicates models with additional pre-training stage. Best: **bold**, second best: underlined.

| Method | | Frames | Workflow | Params (M) | FLOPs (M) | FLOPs/frame (M) | MPJPE↓ |
|---|---|---|---|---|---|---|---|
| PoseFormer Zheng et al. (2021) | ICCV'21 | 27 | Seq2frame | 9.59 | 452 | 452 | 47.0 |
| PoseFormer Zheng et al. (2021) | ICCV'21 | 81 | Seq2frame | 9.60 | 1358 | 1358 | 44.3 |
| MHFormer Li et al. (2022b) | CVPR'22 | 27 | Seq2frame | 18.92 | 1030 | 1030 | 45.9 |
| MHFormer Li et al. (2022b) | CVPR'22 | 81 | Seq2frame | 19.70 | 3132 | 3132 | 44.5 |
| Li et al. Li et al. (2022a) | TMM'22 | 81 | Seq2frame | 4.06 | 392 | 392 | 45.4 |
| Li et al. Li et al. (2022a) | TMM'22 | 243 | Seq2frame | 4.23 | 1372 | 1372 | 44.0 |
| Li et al. Li et al. (2022a) | TMM'22 | 351 | Seq2frame | 4.34 | 2142 | 2142 | 43.7 |
| P-STMO-S Shan et al. (2022) +PT | ECCV'22 | 81 | Seq2frame | 5.4 | 493 | 493 | 44.1 |
| P-STMO Shan et al. (2022) +PT | ECCV'22 | 243 | Seq2frame | 6.7 | 1737 | 1737 | 42.8 |
| Einfalt et al. Einfalt et al. (2023) +PT | WACV'23 | 81 | Seq2frame | 10.36 | 543 | 543 | 45.5 |
| Einfalt et al. Einfalt et al. (2023) +PT | WACV'23 | 351 | Seq2frame | 10.39 | 966 | 966 | 45.0 |
| PoseFormerV2 Zhao et al. (2023) | CVPR'23 | 81 | Seq2frame | 14.35 | 352 | 352 | 46.0 |
| PoseFormerV2 Zhao et al. (2023) | CVPR'23 | 243 | Seq2frame | 14.35 | 1055 | 1055 | 45.2 |
| G-SFormer-S/ +PT | Ours | 81 | Seq2frame | 4.37 | 361 | 361 | 44.1/ 43.5 |
| G-SFormer-S/ +PT | Ours | 243 | Seq2frame | 5.02 | 1092 | 1092 | 42.7/ **41.9** |
| MixSTE Zhang et al. (2022) | CVPR'22 | 81 | Seq2seq | 33.65 | 92692 | 1114 | 42.7 |
| MixSTE Zhang et al. (2022) | CVPR'22 | 243 | Seq2seq | 33.65 | 278076 | 1144 | 40.9 |
| STCFormer Tang et al. (2023) | ICCV'23 | 81 | Seq2seq | 4.75 | 13070 | 13070 | 42.0 |
| STCFormer-L Tang et al. (2023) | ICCV'23 | 243 | Seq2seq | 18.91 | 156392 | 156392 | 40.5 |
| KTPFormer Peng et al. (2024) | CVPR'24 | 81 | Seq2seq | 33.65 | 92706 | 1144 | 41.8 |
| KTPFormer Peng et al. (2024) | CVPR'24 | 243 | Seq2seq | 33.65 | 278119 | 1144 | **40.1** |
| G-SFormer-Encoder/ +PT | Ours | 243 | Seq2seq | 2.25 | 1093 | 4.5 | 43.3/ 42.6 |
| G-SFormer-L-Encoder/ +PT | Ours | 243 | Seq2seq | 4.35 | 2112 | 8.7 | 42.4/ 41.6 |
| G-SFormer/ +PT | Ours | 243 | Seq2frame | 5.54 | 1346 | 1346 | 42.3/ 41.3 |
| G-SFormer-L/ +PT | Ours | 243 | Seq2frame | 7.65 | 2366 | 2366 | 41.6/ 40.5 |

## A.6 DETAILED QUANTITATIVE COMPARISON FOR ROBUSTNESS

In this section, we quantitatively evaluate the robustness of G-SFormer by analyzing its performance under different levels of noise interference. Zero-mean Gaussian noise with varying standard deviations ($\sigma$) is applied to simulate noise of different intensities. Specifically, we randomly select 10% of

the input frames and add noise to two random keypoints in each selected frame. To ensure consistent experimental conditions for a fair comparison, the same random seed is used for all evaluations.

It can be seen from Table 4 that MixSTE (Zhang et al., 2022) suffers significant performance degradation under noise interference, with 10.4 - 24.1mm higher MPJPE compared to the proposed G-SFormer across various noise deviation ranges. PoseFormerV2 (Zhao et al., 2023) shows severe performance deterioration as the noise intensity increases, especially when $\sigma$ exceeds 0.7, culminating in a maximum performance drop of 60.6mm. We also include the noise disturbance results of STCFormer (Tang et al., 2023) which achieves performance comparable to G-SFormer in prior experiments. However, it shows relatively strong performance degradation across the evaluated noise ranges. In contrast, the proposed G-SFormer shows more stable performance trend comprehensively considering both overall accuracy and performance degradation under varying noise intensities. The quantitative results, combined with the qualitative analysis in main manuscript and section A.3, highlight the robustness of G-SFormer against diverse types and levels of noisy input disturbances.

Table 2: Quantitative Comparison with MixSTE (Zhang et al., 2022), PoseFormerV2 (Zhao et al., 2023), and STCFormer(Tang et al., 2023) on Human3.6M dataset of MPJPE (mm). Zero-mean Gaussian noise with varying standard deviations ($\sigma$) is added to random selected frames and keypoints of input 2D poses.

| $\sigma$ | G-SFormer | G-SFormer-L | MixSTE | PoseFormerV2 | STCFormer |
|---|---|---|---|---|---|
| 0 | 42.3 | 41.6 | 40.9 | 45.2 | 42.0 |
| 0.1 | 44.8 | 44.2 | 54.6 | 46.3 | 47.5 |
| 0.2 | 48.8 | 49.4 | 66.3 | 48.9 | 55.2 |
| 0.4 | 55.9 | 54.4 | 78.5 | 59.2 | 70.6 |
| 0.5 | 58.5 | 62.7 | 82.2 | 66.6 | 77.9 |
| 0.7 | 63.5 | 69.3 | 86.9 | 83.5 | 91.1 |
| 0.8 | 65.6 | 72.2 | 88.5 | 91.7 | 97.0 |
| 1.0 | 68.4 | 77.0 | 90.5 | 105.8 | 107.5 |

## A.7 COMPARISON WITH EFFICIENT ATTENTION METHODS

To further verify the effectiveness and efficiency of the proposed Skipped Self-Attention (SSA) in the Skipped Transformer, we incorporate alternative efficient attention mechanisms into the proposed framework as replacements for SSA and compare their performance and computational cost. The base framework used is G-SFormer-S with encoder and decoder layers (L1, L2) set to (3, 5) for 243 frames input. The compared efficient attention methods include: **a. Super Token Attention** in Huang et al. (2022), where self-attention is performed among super tokens, and global-range alignments is built with sparse association. **b. (Shifted) Window Attentions** in Liu et al. (2021; 2022), where self-attention is limited within 3 clips of temporal windows, with cross-window attention achieved through shifted temporal window partitioning in different layers. **c. Max-pooling and Depthwise Conv-pooling attentions** in Fan et al. (2021); Li et al. (2022c), where max-pooling or depthwise convolution pooling reduces temporal length, and pooling strategies for K, Q, and V are decoupled in self-attention computation. Additionally, we include the **standard MHAT in Vanilla Transformer** as the baseline method, which was also introduced in the ablation study (VT-Conv in Table 7) of the main manuscript. Experimental results are listed in Table 3, where "Attn MFLOPs" represents the calculated computational cost of attention mechanism. SSA demonstrates clear superiority in temporal sequence modeling, achieving a balanced trade-off between accuracy and computational efficiency.

Table 3: Comparison of Skipped Self-Attention (SSA) with efficient attention methods

| Method | Attn MFLOPs | MPJPE↓ |
|---|---|---|
| Vanilla MHAT | 30.2 | 44.0 |
| Super Token Attn | 9.3 | 45.7 |
| Window Attn | 10.1 | 44.5 |
| Shifted Window Attn | 10.1 | 45.2 |
| Max-Pooling Attn | 10.2 | 43.3 |
| Depthwise Conv-Pooling Attn | 10.2 | 44.0 |
| SSA (Ours) | 10.1 | **42.7** |

## A.8 INFERENCE SPEED COMPARISON

In this section, we concentrate on the property of Inference Speed and adopt FPS as the evaluation metric, which indicates the number of frames estimated per second. The experimental comparisons are conducted with both $seq2frame$ and $seq2seq$ competitors. We also incorporate the G-SFormer-Encoder to provide the performance in $seq2seq$ workflow. All the evaluations are conducted on a single NVIDIA 3090 GPU, with each model run multiple times over 1000 iterations. The average results are reported for FPS comparison.

Table 4: Computational cost and Inference speed (FPS) comparison with competitive $seq2frame$ and $seq2seq$ methods on Human3.6M. Best: **bold**, second best: underlined.

| Method | Workflow | Params (M) | FLOPs (M) | FLOPs /frame (M) | FPS | GPU Memory (MB) | MPJPE↓ |
|---|---|---|---|---|---|---|---|
| P-STMO Shan et al. (2022) | ECCV'22 | Seq2frame | 6.7 | 1737 | 1737 | 2664 | 11054 | 42.8 |
| PoseFormerV2 Zhao et al. (2023) | CVPR'23 | Seq2frame | 14.35 | 1055 | 1055 | 3872 | 5828 | 45.2 |
| G-SFormer-S | Ours | Seq2frame | 5.02 | 1092 | 1092 | 4231 | 5788 | **41.9** |
| MixSTE-81f Zhang et al. (2022) | CVPR'22 | Seq2seq | 33.65 | 92692 | 1114 | 8895 | 9682 | 42.7 |
| MixSTE Zhang et al. (2022) | CVPR'22 | Seq2seq | 33.65 | 278076 | 1144 | 8883 | 9042 | 40.9 |
| KTPFormer-81f Peng et al. (2024) | CVPR'24 | Seq2seq | 33.65 | 92706 | 1144 | 8445 | 9610 | 41.8 |
| KTPFormer Peng et al. (2024) | CVPR'24 | Seq2seq | 33.65 | 278119 | 1144 | 7935 | 10424 | **40.1** |
| G-SFormer | Ours | Seq2frame | 5.54 | 1346 | 1346 | 3806 | 9056 | 41.3 |
| G-SFormer-L | Ours | Seq2frame | 7.65 | 2366 | 2366 | 2180 | 9162 | 40.5 |
| G-SFormer-Encoder | Ours | Seq2seq | 2.25 | 1093 | 4.5 | 1232284 | 8990 | 42.6 |
| G-SFormer-L-Encoder | Ours | Seq2seq | 4.35 | 2112 | 8.7 | 630617 | 8962 | 41.6 |

As shown in Table 4, G-SFormer-S achieves significantly higher inference speed compared with the efficient $seq2frame$ competitors. While compared with $seq2seq$ methods, the results are even more inspiring. Although the $seq2frame$ workflows fall behind the $seq2seq$ competitors in speed, the $seq2seq$ workflows of G-SFormer/-L mark a substantial improvement. For instance, G-SFormer-Encoder not only delivers higher accuracy than the faster and low-cost version of MixSTE-81f but also achieves an FPS that is **138 times faster**. Similar results can be observed when compared the KTPFormer-81f, G-SFormer-L-Encoder outperforms it with the speed of 630617 FPS, which is **75 times faster**. These experimental results demonstrate that the $seq2sqe$ workflow of G-SFormer provides not only significant reductions in model size and computational cost, but also a remarkable increase in speed, making it an optimal choice for real-world applications involving fast motions.