# OpenReview forum: "Learning Structure-Dynamics-aware Representations for Efficient and Robust 3D Pose Estimation"
_ICLR.cc/2025/Conference — Submitted to ICLR 2025_

### Official Review · Reviewer_Xnmp · 2024-10-16

**Soundness:** 2
**Presentation:** 2
**Contribution:** 2
**Rating:** 6
**Confidence:** 4

**Summary:**

This work proposed G-SFormer, which introduces GNN and Transformer to facilitate 3D human pose estimation. G-SFormer consists of three modules: Spatial Graph Encoder for part-based structural learning within each frame, Skipped Transformer Encoder, and Decoder for hierarchical extraction and aggregation of temporal features. The authors also propose effective data completion methods, which are parameter-free and easy to implement. Experiments are conducted on 3 widely used datasets.

**Strengths:**

1. The method is clearly described
2. Based on the evaluation, the overall quality of the results seems to be satisfactory.

**Weaknesses:**

### 1.Unfair comparison and overclaim.
This is my main concern. The authors claim that the proposed G-SFormer outperforms the previous methods with around 1% computational cost. However, this statement is totally wrong. Given a 2D pose sequence, **the G-SFormer only estimated the 3D pose of the center frame,** which is the seq2frame pipeline. In contrast, **seq2seq methods estimated 3D pose sequence rather than only the center frame** (MixSTE[1], STCFormer[2], MotionBERT[3], MotionAGFormer[4], KTPFormer[5]). To ensure a fair comparison, the authors should evaluate the average computational cost per frame for each method (e.g. **FLOPs/Frames**). As shown in the table, the **seq2seq methods are more efficient than the proposed G-SFormer.** The computation of FLOPs/Frames (or MACs/Frames) is a common evaluation metric in monocular 3D human pose estimation, as demonstrated by previous works such as MotionAGFormer[4], PoseMamba[11] and PoseMagic[12].

|Method|Pipeline|FLOPs (M)|FLOPs/Frames (M)|MPJPE|
|----|----|----|----|----|
|MixSTE[1]|seq2seq|278076|1144|40.9|
|STCFormer[2]|seq2seq|156392 |643|40.5|
|MotionBERT[3]|seq2seq|349434|1438|38.2|
|MotionAGFormer[4]|seq2seq|156492|644|38.4|
|KTPFormer[5]|seq2seq|278119|1144|40.1|
|G-SFormer|seq2frame|2366|2366|40.5|

Moreover, when estimating poses for 243 consecutive frames, **seq2seq methods only require these 243 frames, while seq2frame methods additionally need 121 frames before the left boundary and 121 frames after the right boundary.**

###  2.Limited novelty.
MotionAGFormer[4] combines the GNN and Transformer, achieving better results. While authors have discussed the difference between their G-SFormer and MotionAGFormer (lines 89-90), the MotionAGFormer outperforms the G-SFormer in terms of accuracy and efficiency. We can not simply judge a method as "novel" just because it differs from previous ones; it should also outperform previous methods to be considered truly novel. There are also many works[7-9] that leverage the body parts. In addition, the authors state that "Furthermore, none of them cut to the biggest computational overhead – the Self Attention calculation which is quadratic to the number of tokens. (lines 78-80)" However, HoT[6] has addressed this problem.

### 3.Minor problem.
For optimal image clarity, especially when magnified, it is recommended to use PDF format for all images within the article. PDF format can preserve image quality at various zoom levels.

It is recommended to compare the attention maps with the latest methods instead of P-STMO, which is somewhat outdated. (Appendix A.4)






**Reference**

[1] MixSTE: Seq2seq Mixed Spatio-Temporal Encoder for 3D Human Pose Estimation in Video. CVPR'22

[2] 3D Human Pose Estimation with Spatio-Temporal Criss-cross Attention. CVPR'23

[3] MotionBERT: A unified perspective on learning human motion representations. ICCV'23

[4] MotionAGFormer: Enhancing 3D Human Pose Estimation with a Transformer-GCNFormer Network WACV'24

[5] KTPFormer: Kinematics and Trajectory Prior Knowledge-Enhanced Transformer for 3D Human Pose Estimation CVPR'24

[6] Hourglass Tokenizer for Efficient Transformer-Based 3D Human Pose Estimation CVPR'24

[7] Uncertainty-Aware Human Mesh Recovery from Video by Learning Part-Based 3D Dynamics ICCV'21

[8] Towards Part-aware Monocular 3D Human Pose Estimation: An Architecture Search Approach ECCV'20

[9] Limb Pose Aware Networks for Monocular 3D Pose Estimation IEEE TIP'21

[10] PoseFormerV2: Exploring Frequency Domain for Efficient and Robust 3D Human Pose Estimation CVPR'23

[11] PoseMamba: Monocular 3D Human Pose Estimation with Bidirectional Global-Local Spatio-Temporal State Space Model arXiv'24

[12] Pose Magic: Efficient and Temporally Consistent Human Pose Estimation with a Hybrid Mamba-GCN Network arXiv'24

[13] P-STMO: Pre-Trained Spatial Temporal Many-to-One Model for 3D Human Pose Estimation ECCV'22

**Questions:**

My main concern is the contribution of this work, as discussed in the weakness.

The claim in the title of "..... for efficient and robust 3D pose estimation" is not fully supported by the current manuscript. While the method may offer some advantages, its efficiency is lower than existing seq2seq methods, and the evidence for its robustness is limited. Therefore, I argue that this is not enough to fully justify the core contribution of a research paper in a top-tier conference like ICLR.

---

> ### Author Response · Authors · 2024-11-23
>
> **1. Response for “Unfair comparison and overclaim”**
>
> **(1) Analysis about key evaluation indices**
>
> Given that a model's overall size and computational cost inherently determine its storage and memory requirements, we prioritized these properties when designing G-SFormer. As stated in the manuscript: "the multi-scale G-SFormer models exhibit steady and advanced performance with significantly less computational cost and parameters, making them highly suitable for practical 3D HPE applications on resource-limited platforms."  The model is capable of efficiently processing lengthy input pose sequences while achieving advanced performance with small model size and low computational cost. Major competitors, such as MixSTE(seq2seq), PoseFormerV2(seq2frame), Uplift\&Uplsample(seq2frame), STCFormer(seq2seq), KTPFormer(seq2seq),  also report overall FLOPs in their property analyses, underscoring its importance as a widely acknowledged metric. Our claims regarding low cost and efficiency are based on experimental results.
>
> **Meanwhile, FLOPs/frame is not always suitable for real-world applications.** For example, some downstream tasks like motion prediction use sparse frames at key timesteps as input. In such cases, the dense estimations of seq2seq models become redundant and the large overall computational cost is unnecessary. In contrast, accurately predicting the target frame through low-cost seq2frame workflow is more flexible and efficient.
>
> **(2)Efficiency in the "Seq2seq" Workflow**
>
> However, we also understand the concern of the reviewer about the cost of single frame prediction. Therefore, we present a supplementary Table 1 in Appendix, which organizes the methods using “seq2frame” and “seq2seq” annotations and includes the comparison of FLOPs/frame.
>
> It is worth noting that G-SFormer can also work in the “seq2seq” workflow. As detailed in **Appendix A.5: "DETAILED QUANTITATIVE COMPARISON FOR EFFICIENCY"**, G-SFormer-Encoder achieves an MPJPE of 41.6mm with only 8.7 FLOPs/frame with the full-sequence 3D pose output. This result outperforms low-cost variants of large-scale seq2seq methods while requiring just **0.76% FLOPs for single-frame pose estimation**.
>
> We will add this section of the analysis to the main manuscript to increase objectivity and comprehensiveness.

---

> ### Author Response · Authors · 2024-11-23
>
> **2. Response for "Limited Novelty"**
>
> **(1) Comparison with MotionAGFormer**
>
> For MotionAGFormer (WACV2023), the training and testing data is different from us. The proposed G-SFormer and listed comparing methods are trained and tested with CPN deteced 2D poses with (x,y) keypoint coordinates on Human3.6M. But the given MotionAGFormer employs the input pose data detected by SH, with (x,y, score) as input. Thus both the accuracy of detected 2D poses and input data content are different. We listed the up-to-date SOTA methods including KTPFormer (CVPR2024) which also use CPN detected poses for a fair comparison, and has proved the efficacy by achieving competitive and superior performance on the standard benchmarks.
>
> **(2) Using part-based conception**
>
> Body joints are grouped into 5 parts in spatial modeling of G-SFormer, which is seemed as a frequently-used technique for 3D HPE and other tasks. However, the following key proposals which enable efficient spatial exploitation have not been explored in recent works.
>
> **a.** We deploy Adaptive GNN to construct spatial topology of human body and establish part-wise interactions.
>
> **b.** The coarse-grained part feature, the aggregated graph feature, and the fine-grained joint feature are fused into a comprehensive pose representation.
>
> In contrast, the given comparison methods[7-9] simply employ the physical part concept for feature extraction or constructing separate regression heads.
>
> **(3) Comparison with HoT**
>
> HoT is a plug-and-play module designed to enhance the efficiency of transformers in temporal modeling, partially aligning with the objectives of our work. We have updated the "Introduction" section to include a discussion on HoT, providing a more comprehensive context and enriching our analysis. However, the contribution, solution, deployment, and technical details of our work are significantly different.
>
> **a. Contribution.** The goal of HoT is to "improve existing models", while the proposed G-SFormer is designed as "a practical and standalone model." G-SFormer introduces the Part-based Adaptive GNN and Skipped Transformer to construct a compact architecture for efficient spatial and temporal information processing. In contrast, HoT focuses on reducing temporal tokens in existing models while keeping the spatial domain unchanged. For example, in G-SFormer, joints are fused into a comprehensive pose representation for temporal modeling, whereas HoT retains individual joints for parallel per-joint temporal modeling, which is also a significant computational cost multiplication.
>
> **b. Attention Mechanism.** The Skipped Self-Attention (SSA) in G-SFormer establishes global-range alignments across multiple temporal framesets. Temporal tokens are preserved but built with longer contextual correlations in SSA. Conversely, HoT reduces temporal tokens through KNN-based selection to decrease the computational cost of the Transformer block, which aligns more closely with Uplift\&Upsample and PoseFormerV2. Additionally, the Token Recovering Attention in HoT restores temporal tokens to the original length but does not serve as an efficient attention mechanism, making it technically and functionally distinct from SSA.
>
> In addition, we have detailed the property analysis of SSA compared to other efficient attention mechanisms in Appendix **A.7, COMPARISON WITH EFFICIENT ATTENTION METHODS**, demonstrating its superiority in temporal pose sequence modeling.
>
> **3. Evidence for robustness**
>
>  We have supplemented quantitative robustness comparison in Appendix **A.6 DETAILED QUANTITATIVE COMPARISON FOR ROBUSTNESS.** G-SFormer maintains more stable performance under various noise intensities, further highlighting its robustness against diverse types and levels of noisy input disturbances.

---

> > ### Comment · Reviewer_Xnmp · 2024-11-24
> > **Response to Rebuttal**
> >
> > Thanks to the author for their response.
> >
> > Based on the results in the appendix, G-SFormer's efficiency is good. However, I maintain that the seq2frame pipeline is limited in practical applications. The requirement of a **predefined** target frame, as illustrated in your motion prediction example, is often unrealistic. I still think the novelty of G-SFormer is limited. Moreover, I share similar concerns with Reviewer #pC4k about its robustness.
> >
> > Anyway, I have currently raised the rating to 6 (borderline accept). Either accepting or rejecting is reasonable to me, and I will let the AC make the decision.

---

### Official Review · Reviewer_U2V7 · 2024-10-27

**Soundness:** 3
**Presentation:** 3
**Contribution:** 2
**Rating:** 5
**Confidence:** 3

**Summary:**

This paper proposes a novel method, G-SFormer, which leverages both Transformer and Graph Neural Network (GNN) to improve 3D human pose estimation. The model is composed of three key modules: a Spatial Graph Encoder for part-based structural learning within per frame, and a Skipped Transformer Encoder and Decoder that concurrently establish long-range dynamics for temporal feature extraction across multiple frames. Additionally, the authors introduce parameter-free data completion strategies for 2D pose inputs. The method's effectiveness is demonstrated through extensive experiments on three widely recognized datasets.

**Strengths:**

1. The method’s effectiveness is supported by extensive experiments conducted on multiple widely recognized datasets.
2. The paper introduces a effective approach that addresses key challenges in the field, offering a practical solution and advancing the current state of research.

**Weaknesses:**

1. Unfair comparison.
This is a significant concern. The authors emphasize in the title "… for efficient and robust 3D pose estimation" and assert in the abstract that G-SFormer outperforms previous methods while requiring only "around 1% computational cost". However, this claim is inaccurate. G-SFormer follows a seq2frame approach, estimating the 3D pose only for the center frame in a 2D pose sequence. In contrast, many seq2seq methods, such as MixSTE[1], MotionBERT[2], and KTPFormer[3], estimate the entire 3D pose sequence rather than focusing solely on the center frame. For a fair comparison, the authors should report the average computational cost per frame, such as FLOPs/Frame, a widely used evaluation metric in monocular 3D human pose estimation, as demonstrated by previous works like MotionAGFormer[4].

2. Lack of Innovation in the Proposed Method.
While G-SFormer combines GNN and Transformer to improve 3D human pose estimation, it is not the first method to do so. Previous approaches, such as MotionAGFormer[4], have demonstrated better results, outperforming G-SFormer in both accuracy and computational efficiency. Additionally, the approach of part-based structural learning has already been explored in prior works, such as [5], making it less of a novel contribution in this context. Lastly, the claimed improvement in reducing the computational cost of Transformer-based models is not unique to this paper, as HoT[6] has already addressed this challenge effectively.

Reference
[1] MixSTE: Seq2seq Mixed Spatio-Temporal Encoder for 3D Human Pose Estimation in Video. CVPR'22
[2] MotionBERT: A unified perspective on learning human motion representations. ICCV'23
[3] KTPFormer: Kinematics and Trajectory Prior Knowledge-Enhanced Transformer for 3D Human Pose Estimation CVPR'24
[4] MotionAGFormer: Enhancing 3D Human Pose Estimation with a Transformer-GCNFormer Network WACV'24
[5] Uncertainty-Aware Human Mesh Recovery from Video by Learning Part-Based 3D Dynamics ICCV'21
[6] Hourglass Tokenizer for Efficient Transformer-Based 3D Human Pose Estimation CVPR'24

**Questions:**

Please see more questions in the weaknesses part.

---

> ### Author Response · Authors · 2024-11-23
>
> We sincerely thank you for your valuable suggestions and guidance in improving our work. For detailed solutions and explanations, please refer to our response to reviewer "Xnmp."

---

> > ### Comment · Reviewer_U2V7 · 2024-11-25
> >
> > I can not find detailed comments about the novelty compared with other methods.
> >
> > Lack of Innovation in the Proposed Method. While G-SFormer combines GNN and Transformer to improve 3D human pose estimation, it is not the first method to do so. Previous approaches, such as MotionAGFormer[4], have demonstrated better results, outperforming G-SFormer in both accuracy and computational efficiency. Additionally, the approach of part-based structural learning has already been explored in prior works, such as [5], making it less of a novel contribution in this context. Lastly, the claimed improvement in reducing the computational cost of Transformer-based models is not unique to this paper, as HoT[6] has already addressed this challenge effectively.
> >
> > Please provide comparison results using the FLOPs/Frame protocol.
> > For a fair comparison, the authors should report the average computational cost per frame, such as FLOPs/Frame, a widely used evaluation metric in monocular 3D human pose estimation, as demonstrated by previous works like MotionAGFormer[4].

---

> > > ### Author Response · Authors · 2024-11-26
> > >
> > > We apologize for not addressing your concerns in a timely manner. Here, we provide explanations and clarifications on the key issues.
> > >
> > > **1. Response for "Lack of Innovation"**
> > >
> > > **(1) Comparison with MotionAGFormer**
> > >
> > > Although both the given MotionAGFormer and the proposed G-SFormer contain the concept of GNN and Transformer, their design principles and functions are fundamentally different.
> > >
> > > **a. Integrated Framework Design**
> > >
> > > As discussed in line 102-106 in Introduction, *"Different from existing GraphTransformer hybrid methods (Zhu et al., 2021; Zhao et al., 2022; Soroush Mehraban, 2024) that embed GCN into Transformer block to assist self-attention in spatial modeling, G-SFormer integrates part-based adaptive GNN and Skipped Transformer to efficiently exploit spatial and temporal information, respectively."*
> > >
> > > MotionAGFormer utilizes the adjacency matrix in GCN to build join-wise and frame-wise alignments, thus replacing the self-attention in Transformer. In contrast, GNN and Transformer in G-SFormer operate independently for spatial and temporal modeling. A concise graph-encoded pose representation integrated with multi-granularity pose attributes is delivered to the newly proposed Skipped Transformer, without the redundant and repetitive spatio-temporal connections in dual-stream MotionAGFormer. This streamlined framework enables efficient modeling of the lengthy input pose sequence.
> > >
> > > **b. Adaptive Graph Topoloty based on Body Parts**
> > >
> > > G-SFormer introduces a novel Part-based Adaptive GNN with a topology adaptively learned through graph attention mechanism, rather than a predefined joint-wise topology used in MotionAGFormer, with its superiority regarding model efficiency and flexibility is discussed in line 090-096 in Introduction:
> > >
> > > *"Unlike present methods which compute joint-wise connections (Zhang et al., 2022; Tang et al., 2023; Yu et al., 2023; Peng et al., 2024), our approach builds spatial correlations among body parts to better represent the coordination of human body and the interaction between body parts during movement. For instance, the arms and torso are closely related during ”Eating”, while the legs are correlated for ”Sitting”. The part-based graph structure is fully adaptive, learned through a graph attention mechanism without relying on pre-defined skeletal topology as priors (Soroush Mehraban, 2024; Yu et al., 2023; Penget al., 2024), thereby enhancing model flexibility and generalization across diverse poses."*
> > >
> > > **c. Efficacy of G-SFormer**
> > >
> > > For MotionAGFormer (WACV2023), the training and testing data is different from us. The proposed G-SFormer and listed comparing methods are trained and tested with CPN detected 2D poses with (x,y) keypoint coordinates on Human3.6M. But the given MotionAGFormer employs the input pose data detected by SH, with (x,y, score) as inputs. Thus both the accuracy of detected 2D poses and input data content are different. We listed the up-to-date SOTA methods including KTPFormer (CVPR2024) which also use CPN detected poses for a fair comparison, and has proved the efficacy by achieving competitive and superior performance with significant smaller model size and lower cost on the standard benchmarks.
> > >
> > > **(2)  Using part-based conception**
> > >
> > > Body joints are grouped into 5 parts in spatial modeling of G-SFormer, which is seemed as a frequently-used technique for 3D HPE and other tasks. However, the following key proposals which enable efficient spatial exploitation have not been explored in recent works.
> > >
> > > **a.** We deploy Adaptive GNN to construct spatial topology of human body and establish part-wise interactions.
> > >
> > > **b.** The coarse-grained part feature, the aggregated graph feature, and the fine-grained joint feature are fused into a comprehensive pose representation.
> > >
> > > In contrast, the given comparison methods[5] simply employs the physical part concept for feature grouping and constructing separate regression heads.

---

> > > ### Author Response · Authors · 2024-11-26
> > >
> > > **(3) HoT comparison**
> > >
> > > HoT is a plug-and-play module designed to enhance the efficiency of transformers in temporal modeling, partially aligning with the objectives of our work. We have updated the "Introduction" section to include a discussion on HoT, providing a more comprehensive context and enriching our analysis. However, the contribution, solution, deployment, and technical details of our work are significantly different.
> > >
> > > **a. Contribution.** The goal of HoT is to "improve existing models", while the proposed G-SFormer is designed as "a practical and standalone model." G-SFormer introduces the Part-based Adaptive GNN and Skipped Transformer to construct a compact architecture for efficient spatial and temporal information processing. In contrast, HoT focuses on reducing temporal tokens in existing models while keeping the spatial domain unchanged. For example, in G-SFormer, joints are fused into a comprehensive pose representation for temporal modeling, whereas HoT retains individual joints for parallel per-joint temporal modeling, which is also a significant computational cost multiplication.
> > >
> > > **b. Attention Mechanism.** The Skipped Self-Attention (SSA) in G-SFormer establishes global-range alignments across multiple temporal framesets. Temporal tokens are preserved but built with longer contextual correlations in SSA. Conversely, HoT reduces temporal tokens through KNN-based selection to decrease the computational cost of the Transformer block, which aligns more closely with Uplift\&Upsample and PoseFormerV2. Additionally, the Token Recovering Attention in HoT restores temporal tokens to the original length but does not serve as an efficient attention mechanism, making it technically and functionally distinct from SSA.
> > >
> > > In addition, we have detailed the property analysis of SSA compared to other efficient attention mechanisms in **Appendix A.7, COMPARISON WITH EFFICIENT ATTENTION METHODS**, demonstrating its superiority in temporal pose sequence modeling.
> > >
> > >
> > > **2. Response for Per-frame Prediction Cost Analysis**
> > >
> > > Thank you for your suggestion to include the FLOPs/frame metric, which provides a more comprehensive analysis of the performance. We have presented a supplementary table in Appendix, which presents the methods more intuitive with “seq2frame” and “seq2seq” annotations and analyzes the computation cost for single frame prediction.
> > >
> > > It is worth noting that G-SFormer can also work in the “seq2seq” workflow. As detailed in **Appendix A.5: "DETAILED QUANTITATIVE COMPARISON FOR EFFICIENCY"**, G-SFormer-Encoder achieves an MPJPE of 41.6mm with only 8.7 FLOPs/frame with the full-sequence 3D pose output. This result outperforms low-cost variants of large-scale seq2seq methods while requiring just **0.76% FLOPs for single-frame pose estimation**.
> > >
> > > We will add this section of the analysis to the main manuscript to increase objectivity and comprehensiveness.

---

### Official Review · Reviewer_2qgK · 2024-10-29

**Soundness:** 3
**Presentation:** 3
**Contribution:** 3
**Rating:** 5
**Confidence:** 5

**Summary:**

This paper proposes a simple yet effective Graph and Skipped Transformer (G-SFormer) for 3D human pose estimation. It leverages a Part-based Adaptive GNN and a Frameset-based Skipped Transformer to capture both detailed pose representations and multi-perspective dynamic representations. Experimental results show that G-SFormer achieves competitive performance on benchmark datasets, including Human3.6M, MPI-INF-3DHP, and HumanEva.

**Strengths:**

The paper is well-structured and clearly presented.
The idea is simple, intuitive yet effective.

**Weaknesses:**

- The proposed method does not compare with the state-of-the-art method [1,2]. For example, MotionAGFormer achives 38.4 mm in MPJPE on Human3.6M, which achive better performance than the proposed method.
- Although the authors claim that their method can address joint noise, no clear evidence or quantitative results are provided to substantiate this claim. It is recommended that the authors include experimental comparisons and present evidence demonstrating why the proposed method effectively addresses joint noise to strengthen their argument.
- Given the paper’s emphasis on efficiency, a time-comparison table (inference time) should be included, as a FLOPs comparison alone is insufficient. Furthermore, since G-SFormer only regresses the 3D pose of the center frame, it appears to achieve a much lower speed compared to seq2seq-based methods [1,2,3].
- The paper lacks a comparison with other efficient methods, while I think it is necessary. Such as flash-attention [4] and state-space models [5].

[1] MotionAGFormer: Enhancing 3D Human Pose Estimation with a Transformer-GCNFormer Network

[2] MotionBERT: Unified Pretraining for Human Motion Analysis

[3] MixSTE: Seq2seq Mixed Spatio-Temporal Encoder for 3D Human Pose Estimation in Video

[4] FlashAttention: Fast and Memory-Efficient Exact Attention with IO-Awareness

[5] Mamba: Linear-Time Sequence Modeling with Selective State Spaces

**Questions:**

See Weaknesses

---

> ### Author Response · Authors · 2024-11-23
>
> **1. Comparison methods**
>
> For MotionAGFormer (WACV2023) and MotionBert (ICCV2023), the training and testing data are different from us. The proposed G-SFormer and listed comparing methods are trained and tested with CPN deteced 2D poses with (x,y) keypoint coordinates on Human3.6M. While the given MotionAGFormer and MotionBert employ the input pose data detected by SH, with (x,y, score) as input. Thus both the accuracy of detected 2D poses and input data content are different. We listed the up-to-date SOTA methods including KTPFormer (CVPR2024) which also use CPN detected poses for a fair comparison and property analysis.
>
> **2. Evidence for Robustness**
>
> Thank you for your suggestion. We have noticed the deficiency and conducted additional experiments to further verify the robustness of G-SFormer with quantitative study. Please check **A.6 DETAILED QUANTITATIVE COMPARISON FOR ROBUSTNESS** in Appendix for more detail. G-SFormer maintains a more stable performance trend under various noise intensities, further highlighting its robustness against diverse types and levels of noisy input disturbances.
>
> **3. Efficiency concern**
>
> For most of the comparison methods have not given the FPS property, it is currently hard for us to present a time-cost comparison and analysis. To address this, we have included the computational cost for single frame pose estimation in **Appendix A.5: "DETAILED QUANTITATIVE COMPARISON FOR EFFICIENCY"** as an auxiliary indicator, and made more intuitive illustrations with “seq2seq” and “seq2frame” annotations.
>
> Compared with methods with “seq2frame” workflow, G-SFormer-S can outperform all the competitors, the model size and computational cost are comparable to the efficient HPE model PoseformerV2, but achieves significantly higher accuracy.
>
> We agree with the reviewer's observation that the "seq2frame" workflow struggles to match "seq2seq" models in terms of per-frame pose generation cost. However, we would like to emphasize two key aspects that highlight the efficiency of the proposed G-SFormer:
>
> **(1) Low Requirements for Storage and Computational Memory.** As stated in the Introduction, "the multi-scale G-SFormer models exhibit steady and advanced performance with significantly lower computational cost and fewer parameters, making them highly suitable for practical 3D HPE applications on resource-limited platforms." G-SFormer employs compact designs including the Part-based Adaptive GNN and Skipped Transformer to efficiently process spatial and temporal information in lengthy input pose sequences, achieving competitive and superior performance compared to both "seq2seq" and "seq2frame" approaches.
>
> **(2) Efficiency in the "Seq2seq" Workflow.** G-SFormer can also operate in the "seq2seq" workflow with significantly lower FLOPs/frame while delivering advanced performance. In Appendix A.5, we present the results of the full-sequence 3D poses generated by the G-SFormer-Encoder, achieving an MPJPE of 41.6mm with only 8.7 FLOPs/frame. This performance surpasses the low-cost variants of large-scale seq2seq methods while requiring just **0.76% of the FLOPs for single-frame pose estimation.**
>
> **4. Comparison with efficient methods**
>
> Due to the limited rebuttal time, it is currently hard for us to provide experimental results of the variants using FlashAttention and Mamba as replacements for the proposed Skipped Self-Attention (SSA). While we attempted to incorporate the Mamba-embedded Transformer, we have not yet achieved optimized results despite multiple structural adjustments and parameter fine-tuning. However, we have conducted the experiments to compare SSA with other efficient attention mechanisms, with results presented in Appendix **A.7, COMPARISON WITH EFFICIENT ATTENTION METHODS**. The proposed SSA shows superiority considering both accuracy and computational cost in temporal pose sequence modeling.

---

> > ### Comment · Reviewer_2qgK · 2024-11-26
> >
> > Thank you for your reply. The rebuttal did not address my concerns. It avoids comparing the proposed method with other SOTA approaches, which I find unfair. While it claims efficiency, it fails to provide comparisons with other efficient methods in terms of FPS. A comprehensive comparison with SOTA methods is essential to evaluate the trade-offs between accuracy and inference time. As it stands, the experiments do not convincingly demonstrate the effectiveness or efficiency of the proposed method. I have also reviewed the other reviews, and considering the novelty and the experimental results, I don’t believe it meets the threshold for acceptance at ICLR.

---

> > > ### Author Response · Authors · 2024-12-01
> > >
> > > We regret that our previous rebuttal did not address your concerns. Here, we provide further clarification on the key issues.
> > >
> > > **1. Efficiency in FPS**
> > >
> > > We have supplemented the FPS comparison experiments to further evaluate the speed of G-SFormer. Experiments are conducted across both seq2frame and seq2seq workflows using challenging and representative competitors. Detailed experimental settings and results have been supplemented in **Appendix A.8: INFERENCE SPEED COMPARISON**. Here we provide a concise version of **Table 4** from the Appendix for reference.
> > >
> > > | **Method**             | **Workflow** | **Params** | **FLOPs/frame** | **FPS** | **GPU Memory** | **MPJPE** |
> > > |:----------------------|:------------|:----------:|:---------:|:-------:|:---------------:|:---------:|
> > > | P-STMO (ECCV'22)       | Seq2frame    | 6.7        | 1737      | 2664    | 11054           | 42.8      |
> > > | PoseFormerV2 (CVPR'23) | Seq2frame    | 14.35      | 1055      | 3872    | 5828            | 45.2      |
> > > | G-SFormer-S (ours)     | Seq2frame    | 5.02       | 1092      | 4231    | 5788            | 41.9      |
> > >
> > > | **Method**                 | **Workflow** | **Params** | **FLOPs/frame** | **FPS** | **GPU Memory** | **MPJPE** |
> > > |:--------------------------|:------------|:----------:|:---------------:|:-------:|:---------------:|:---------:|
> > > | MixSTE-81f (CVPR'22)       | Seq2seq      | 33.65      | 1114            | 8895    | 9682            | 42.7      |
> > > | MixSTE (CVPR'22)           | Seq2seq      | 33.65      | 1144            | 8883    | 9042            | 40.9      |
> > > | KTPFormer-81f (CVPR'24)    | Seq2seq      | 33.65      | 1144            | 8445    | 9610            | 41.8      |
> > > | KTPFormer (CVPR'24)        | Seq2seq      | 33.65      | 1144            | 7935    | 10424           | 40.1      |
> > > | G-SFormer (ours)           | Seq2frame    | 5.54       | 1346            | 3806    | 9056            | 41.3      |
> > > | G-SFormer-L (ours)         | Seq2frame    | 7.65       | 2366            | 2180    | 9162            | 40.5      |
> > > | G-SFormer-Encoder (ours)   | Seq2seq      | 2.25       | 4.5             | 1232284 | 8990            | 42.6      |
> > > | G-SFormer-L-Encoder (ours) | Seq2seq      | 4.35       | 8.7             | 630617  | 8962            | 41.6      |
> > >
> > > These experimental results demonstrate that the seq2sqe workflow of G-SFormer provides not only significant reductions in model size and computational cost, but also a remarkable increase in speed, which is **75/138 times faster** compared with the competitive seq2seq methods. The superiority in speed further makes it an optimal choice for real-world applications involving fast motions.
> > >
> > > **2. Comparison Methods**
> > >
> > > We understand the reviwer's concern. Here, we would like to further clarify our perspective and reasoning.
> > >
> > > **(1)** We have already compared our method with KTPFormer (CVPR'24), which is an up-to-date and strong baseline that demonstrates outstanding performance on standard large-to-small benchmarks, using the same experimental data as us.
> > > Given the inherent differences in the 2D pose input in Human3.6M dataset, and considering that our efficiency-speed-accuracy comparison is based on this dataset, we hold the view that the choice of competitive methods is reasonable.
> > >
> > > **(2)** Considering the key efficiency metrics including FPS, FLOPs/frame, and overall FLOPs, the superior performance of G-SFormer in efficiency is supported by the experimental results and demonstrates significant advantages. Meanwhile, refering the observations and comments of Reviewer Xnmp and Reviewer kpzv, the advanced performance in efficiency is sufficient to demonstrate the method's efficacy.
> > >
> > > We hope our statement and clarification will be helpful. Thank you once again for your time and consideration.

---

### Official Review · Reviewer_Epe9 · 2024-11-03

**Soundness:** 2
**Presentation:** 3
**Contribution:** 1
**Rating:** 3
**Confidence:** 4

**Summary:**

This paper mainly focuses on accelerating the temporal 2D-3D human pose lifting task. The authors propose a Part-based Adaptive Graph Neural Network (GNN) to dynamically model joints more robustly and efficiently. Additionally, they introduce a skipped transformer to handle temporal redundancy more effectively. Comparisons with other state-of-the-art methods demonstrate its computational efficiency while maintaining good performance.

**Strengths:**

- The writing of this paper is well-organized and easy to understand.

**Weaknesses:**

- Vector graphics should be used.
- In Figure 1(a), the number of frames used by each model should be annotated at each point for a clearer and fairer comparison.
- Line 78 states, "none of them cut to the biggest computational overhead – the Self Attention calculation which is quadratic to the number of tokens." The improvements to attention here primarily involve modifications to the input length; Deciwatch [1] also makes similar improvements and should be compared.
- The related work section should also introduce efficient human pose lifting.
- The Part-based Adaptive GNN dynamically aggregates part embedding after partitioning the human body into parts, which is similar to the MTF-Transformer [2]. The innovation needs to be clarified, and a simple comparative experiment should be conducted.
- Line 93-94 states, "without relying on pre-defined skeletal topology as priors," but the partitioning of different body parts inherently includes priors about the human body. Difference between the weights of constructed human part graphs and human skeletal structure does not support this point. If feasible, randomly partition the body to evaluate the model in the absence of any human priors.
- The method of partitioning the body is not well explored in the paper. For example, how would the results change if each joint is treated as an independent part? How would the results change if there are overlapping joints between the parts?
- Table 5 lacks experiments for w/o SPE + w/o DR and comparisons with raw data expanding (Figure 4(c)).
- Table 6 lacks comparisons for m=1 regarding the effects of Spatial-MLP and joint-wise GCN, as the influence of skip attention should be excluded. Additionally, including experiments with a pure transformer to model joints for comparison would be beneficial to comprehensive comparison.

[1] DeciWatch: A Simple Baseline for 10× Efficient 2D and 3D Pose Estimation

[2] Adaptive Multi-View and Temporal Fusing  Transformer for 3D Human Pose Estimation

**Questions:**

- How exactly is AMASS utilized? The paper does not explain this in detail. AMASS is a significantly larger dataset than Human 3.6M, but the improvement after pre-training with AMASS is generally moderate; this section lacks analysis.
- Regarding data padding, should at most T/2 frames be padded for a sequence of T frames?
- The paper uses SPE for PE(position encoding). What advantages do SPE have over the commonly used learnable PE? Compare visualizations of the two position encodings, showing cosine similarity between positions, would be helpful. The benefits of encoding relative positional information for temporal or joint modeling are not addressed. If this is important, why not use the rotary PE commonly used in LLMs?

---

> ### Author Response · Authors · 2024-11-23
>
> 1 Since vector graphics are not mandatory for the submission, they should be considered as a recommendation rather than a requirement.
>
> 2. DeciWatch[1] is a 2D/3D pose estimation method with RGB Video Frames as input and MTF-Transformer[2] is a Multi-view pose estimation model. Neither of these methods falls within the domain of single-view 2D-to-3D lifting-based pose estimation, which makes a direct comparison infeasible.

---

> > ### Comment · Reviewer_Epe9 · 2024-11-26
> >
> > Both MTF-Transformer and Deciwatch share similarities with the approach in this paper, despite differences in their specific domains. Their overall pipelines are quite similar to the method proposed here.
> > -  **MTF-Transformer** adopts part-based structural learning, which is view-irrelvant and temporal-irrelvant. We would like the authors to elaborate on the similarities and differences between part-based methods (such as MTF-Transformer) and the approach in this paper, and preferably compare their performance differences (at least two methods). This will allow us to better evaluate the novelty of this paper.
> > - **Deciwatch**, although using RGB frames as input, similarly enhances performance by compressing temporal sequence lengths. It was the first method in its field to adopt this approach and retains the ability to reconstruct the full sequence of frames. We recommend that the authors explore additional temporal keyframe clustering methods, rather than relying solely on fixed step intervals—such as clustering based on attention scores.

---

> ### Author Response · Authors · 2024-12-01
>
> We understand the reviewers' concerns regarding the spatial and temporal information processing method of G-SFormer. Here, we provide clarification on the structural designs.
>
> **1.Compare with Part-based design in MTF-Transformer**
>
> G-SFormer employs the part-based design to better represent the coordination of human body and the interaction between body parts during movement. The key proposals which enable adaptive and efficient spatial exploitation are as follows:
>
> **a.** We deploy a fully data-driven Adaptive GNN to construct spatial topology of human body and establish part-wise interactions.
>
> **b.** The coarse-grained part feature, the aggregated graph feature, and the fine-grained joint feature are fused into a comprehensive pose representation.
>
> In contrast, the MTF-Transformer simply employs the physical part concept for separately feature extraction.
>
> **2. Compare with Temporal Compression method in Deciwatch**
>
> Deciwatch adopts uniformly sampling to the input RGB video frames to reduce the overall computational cost and improve efficiency. It is a lossy temporal compression method which is fundamentally different from ours.  The proposed Skipped Transformer achieves efficient temporal modeling without discarding temporal tokens, thus preserving the integrity of the information. Instead, it employs Skipped Self-Attention (SSA) to enhance efficiency and capture long-range dynamics across multiple framesets, enabling advance performance while significantly reducing computational cost.
>
> **3. Efficient-Attention Comparison**
>
> We have conducted experiments to compare SSA with other efficient attention mechanisms, including the Pooling Attention, Window Attention, Super Token Attention, etc. The Super Token Attention incorporates the concept of clustering through sparse associations among super tokens. Please refer to the results presented in **Appendix A.7, COMPARISON WITH EFFICIENT ATTENTION METHODS** for detail. The proposed SSA shows superiority considering both accuracy and computational cost in temporal pose sequence modeling.

---

### Official Review · Reviewer_kpzv · 2024-11-03

**Soundness:** 2
**Presentation:** 2
**Contribution:** 3
**Rating:** 5
**Confidence:** 3

**Summary:**

This work proposes a method for 2D-to-3D human pose sequence lifting. The focus is on efficiency. The method fits into a long line of research using spatiotemporal transformers that aggregate information both across time and space (meaning here the human body parts).
The novelty lies in using a "Frameset-based Skipped Transformer" for the temporal aspect, and a Part-based Adaptive Graph Neural Network for the spatial aspect. The former achieves significant improvements in efficiency by not computing attention between all pairs of frames. The latter allows learning graph connections from data, instead of hand-specifying them, e.g. based on the usual skeletal structure. Furthermore a new strategy for padding input poses at the edges of the sequence is also proposed.
The method achieves or approaches the state-of-the-art on Human3.6M and MPI-INF-3DHP, while having much lower computational cost as measured in FLOPs. Ablations on Human3.6M show that the proposed techniques are effective.

**Strengths:**

* Temporal human pose lifting from 2D to 3D is a topic with a lot of research interest in recent years and research into making these models more efficient will have positive impact in this particular community.
* The method achieves substantially lower computational cost compared to prior works at similar levels of joint error.
* Models of different sizes are proposed for tradeoff selection between speed and accuracy.
* The ablations verify that the skipped transformer and the adaptive GNN are bringing benefits.
* The presentation of the related works is comprehensive and comparisons are done with the latest works in this area.

**Weaknesses:**

* Only studio datasets are used. Performing experiments on datasets such as 3DPW or EMDB would strengthen the claims.
* The proposed data completion methods, as well as the sinusoidal positional encodings, bring tiny improvements only, which might be due to noise or might not generalize to other datasets. (Table 3.).
* Since the focus is on efficiency, actuall wall-clock inference time comparison would also be important, not only FLOPs and parameter counts. See e.g. [1].
* The presentation could be improved. Font sizes in the figures are unreadably small. Similarly with Tables 3 and 4.
* The contribution could be called incremental, since reduced-cost attention variants are a very-well researched area. The impact is therefore narrower and limited to the 2D-to-3D pose lifting community, for which a better venue might be CVPR/ICCV.

[1] Dehghani et al. The Efficiency Misnomer. ICLR 2022

**Questions:**

Minor suggestions:

Eq. 4. would be clearer with explicitly saying "concat" or "cat" or using || as an infix operator.

L102 typo "sate of the art"
L262 ".:"
L350: matrics -> metrics
It is typically better to place all tables and figures to the top of the page (LaTeX [t])

---

> ### Author Response · Authors · 2024-11-23
>
> **1. Dataset Selection**
>
> Currently, the comparison lifting-based pose estimation methods are based upon the datasets including Human3.6M, 3DHP and HumanEva. Therefore, we chose the same datasets for performance evaluation. We agree with the reviewer that the wild dataset could further strengthen the claim, and will conduct experiments in future study.
>
> **2.  Data Completion Methods**
>
> We have updated Table 5 in main manuscript to include the condition where neither SPE nor DR is used. Please refer to the updated table for a clearer comparison. The improvements introduced by data completion methods were evaluated through extensive experiments using G-SFormer models of varying scales (G-SFormer/-S/-L) during training and testing.
>
> **3. Efficiency Comparison**
>
> Since most comparison methods have not report FPS properties, conducting a complete property comparison and analysis is challenging within the limited rebuttal time. To address this, we have included the computational cost for single frame pose estimation (FLOPs/frame) in **Appendix A.5: "DETAILED QUANTITATIVE COMPARISON FOR EFFICIENCY"** as an auxiliary indicator. This section details the overall and single-frame pose estimation cost of G-SFormer in both seq2seq and seq2frame workflows.
>
> **4. Manuscript Revision**
>
> Thank you for your suggestion. We have corrected the writing errors and are actively revising our manuscript to enhance its overall clarity, visualization, and presentation.
>
> **5. Skipped Self-Attention Efficacy**
>
> In Appendix **A.7: "COMPARISON WITH EFFICIENT ATTENTION METHODS"**, we have supplemented the experiments to compare the proposed Skipped Self-Attention with other efficient attention methods, demonstrating its superiority in terms of computational cost and accuracy for temporal sequence modeling.

---

> > ### Comment · Reviewer_kpzv · 2024-11-26
> >
> > The updates do help in clarifying the issues. The discussion of seq2frame vs seq2seq with the other reviewer are also important, and the new table is an improvement in this regard. I still think that an efficiency-focused paper should provide timings measured on real hardware, since efficiencies in FLOPs do not directly translate to speed gains. I therefore still see this as a marginal and incremental work, slightly above the threshold.

---

> > > ### Author Response · Authors · 2024-11-27
> > >
> > > Thank you for taking the time to review our supplemented material. Benefiting from the extended discussion period, we had additional time to conduct experiments and completed the FPS comparison, a crucial metric for evaluating the speed of the proposed model. Experiments are conducted across both seq2frame and seq2seq workflows using challenging and representative competitors. Detailed experimental settings and results have been supplemented in **Appendix A.8: INFERENCE SPEED COMPARISON**. Here we provide a concise version of **Table 4** from the Appendix for reference.
> > >
> > > | **Method**             | **Workflow** | **Params** | **FLOPs/frame** | **FPS** | **GPU Memory** | **MPJPE** |
> > > |:----------------------|:------------|:----------:|:---------:|:-------:|:---------------:|:---------:|
> > > | P-STMO (ECCV'22)       | Seq2frame    | 6.7        | 1737      | 2664    | 11054           | 42.8      |
> > > | PoseFormerV2 (CVPR'23) | Seq2frame    | 14.35      | 1055      | 3872    | 5828            | 45.2      |
> > > | G-SFormer-S (ours)     | Seq2frame    | 5.02       | 1092      | 4231    | 5788            | 41.9      |
> > >
> > > | **Method**                 | **Workflow** | **Params** | **FLOPs/frame** | **FPS** | **GPU Memory** | **MPJPE** |
> > > |:--------------------------|:------------|:----------:|:---------------:|:-------:|:---------------:|:---------:|
> > > | MixSTE-81f (CVPR'22)       | Seq2seq      | 33.65      | 1114            | 8895    | 9682            | 42.7      |
> > > | MixSTE (CVPR'22)           | Seq2seq      | 33.65      | 1144            | 8883    | 9042            | 40.9      |
> > > | KTPFormer-81f (CVPR'24)    | Seq2seq      | 33.65      | 1144            | 8445    | 9610            | 41.8      |
> > > | KTPFormer (CVPR'24)        | Seq2seq      | 33.65      | 1144            | 7935    | 10424           | 40.1      |
> > > | G-SFormer (ours)           | Seq2frame    | 5.54       | 1346            | 3806    | 9056            | 41.3      |
> > > | G-SFormer-L (ours)         | Seq2frame    | 7.65       | 2366            | 2180    | 9162            | 40.5      |
> > > | G-SFormer-Encoder (ours)   | Seq2seq      | 2.25       | 4.5             | 1232284 | 8990            | 42.6      |
> > > | G-SFormer-L-Encoder (ours) | Seq2seq      | 4.35       | 8.7             | 630617  | 8962            | 41.6      |
> > >
> > > These experimental results demonstrate that the seq2sqe workflow of G-SFormer provides not only significant reductions in model size and computational cost, but also a remarkable increase in speed, which is **75/138 times faster** compared with the competitive seq2seq methods. The superiority in speed further makes it an optimal choice for real-world applications involving fast motions.

---

> > > > ### Comment · Reviewer_kpzv · 2024-11-27
> > > >
> > > > These numbers are impressive, though I'm not sure where this 138x speedup (over e.g., MixSTE in seq2seq mode) really comes from. The skipped transformer with an m=3 should bring a factor of 3x reduction over the full attention.
> > > >
> > > > My impression is that the paper is not well-structured for understanding the core contribution and feels somewhat disorganized. If the method is so much more efficient, then the paper should present the reason in a much more clear way, instead of also mixing in aspects of joint noise, learned spatial graph connections etc.
> > > > In fact, upon more reflection, I think it could be beneficial to rework the paper just concentrating on efficiency, and describing it clearly where exactly this efficiency is gained.

---

> > > > > ### Author Response · Authors · 2024-12-01
> > > > >
> > > > > Thank you for your specific suggestions regarding the strengths and weaknesses of our work. Please allow us to further clarify some key issues.
> > > > >
> > > > > **1.** The description of “138 times speedup” comes from the comparison of G-SFormer-Encoder and the low-cost version of MixSTE-81f regarding the FPS performance.
> > > > >
> > > > > **2.** As the reviewer observed, the Skipped Self-Attention with m=3 brings a factor of 3x reduction over the full attention. However, it is one of the reason for the low-cost and high-efficiency of the proposed G-SFormer, another important reason is the streamlined framework that efficiently exploits spatial and temporal information in the input pose sequence.
> > > > >
> > > > > **(1) In spatial modeling**, G-SFormer adopts coarse-grained body-parts to build spatial correlations, and combined with multi-granularity pose attributes to generate one compact pose representation for temporal modeling.
> > > > >
> > > > > **(2) In temporal modeling**, G-SFormer adopts Skipped Self-Attention among framesets to establish global-range alignments across multiple temporal perspectives.
> > > > >
> > > > > In contrast, the mainstream methods typically employ dense joint-frame connections and iterative correlations, which introduce large computational redundancy. As illustrated in **Line 048-052**:
> > > > >
> > > > > “*Prior methods typically deploy self-attention to establish joint-wise correlations, as well as frame-wise correlations for each joint individually or for the encoded pose representation (Zhang et al., 2022; Zheng et al., 2021). This is computationally expensive especially when dealing with lengthy sequences (81, 243 or even more), rendering it impractical for deployment on resource-limited mobile devices and consumer hardware.*”
> > > > >
> > > > > Therefore, the efficient design of the overall framework, including the Part-based Adaptive GNN and Skipped Transformer mutually contribute to the high efficiency of G-SFormer.

---

> > > > > > ### Comment · Reviewer_kpzv · 2024-12-01
> > > > > >
> > > > > > I thought that the discussion period ended on Nov 26, but apparently it's still possible to write comments.
> > > > > >
> > > > > > I'm still confused about which design choices, and how, exactly cause the 138-fold speedup over MixSTE. If the main contribution of the paper is efficiency, this has to be clearly explained with calculations. The Skipped Self-Attention gives a factor of 3x. But 138/3=46. Where does the 46x speedup come from? It seems most of the efficiency comes from the spatial modeling changes? If this is the case, then the paper should include the calculations on this. Currently, only the Skipped Self-Attention's computational complexity is discussed in Sec. 3.2.1. The introduction also does not mention that the efficiency gains are from the spatial modeling changes. It says the problem is long temporal sequences and dense temporal attention. But it now seems that a bigger issue with prior works is that they use joint-wise tokens? It makes the paper harder to read that several aspects are mixed, such as noise robustness, efficiency, learned graph structure for the spatial modeling. I find it difficult to distill an overall message. I would suggest to focus on a core contribution. If the efficiency gains are as big as suggested in the new tables added for rebuttal, this can be a strong contribution and can make a big impact. But the current writeup is too confusing for this.
> > > > > >
> > > > > > For example, the impression from Fig. 1a is not such a large gain in FLOPs. The gain over P-STMO is not so extreme. Then Fig. 1b concentrates on errors like joint swaps, but if the paper is focused on efficiency, this is not what the main message should be, it distracts from the contribution. Also, MixSTE is compared to in Fig. 1b, but not plotted in Fig. 1a. I think the paper would benefit from a reworking to deliver a coherent message.

---

### Official Review · Reviewer_pC4k · 2024-11-04

**Soundness:** 3
**Presentation:** 3
**Contribution:** 2
**Rating:** 5
**Confidence:** 4

**Summary:**

This paper proposes G-SFormer, a combination of a part-based GNN and a skipped transformer for 3D human pose estimation. The proposed approach achieves competitive results on three datasets while being efficient due to the frame-skipping strategy. Comprehensive experiments and ablation studies are presented to verify the effectiveness of the approach.

**Strengths:**

1. This paper shows comprehensive experiment results and ablation studies.
2. The proposed approach, G-SFormer, achieves competitive performance on standard benchmarks while being efficient compared to state-of-the-art methods.

**Weaknesses:**

1. The proposed method combines a part-based GNN for frame-feature extraction with a skipped transformer for efficient temporal processing, which seems incremental.
2. While it's easy to understand that the proposed model architecture is efficient due to the lower cost of temporal self-attention, it remains unclear to me why the learned representation is robust against noisy 2D keypoints. The model architecture does not appear to be well-motivated with respect to this goal. Can authors provide insights regarding this? In addition to the qualitative results shown in the paper, I suggest the authors also present quantitative results like Fig. 6 of PoseFormerV2 [1], where Gaussian noise is added to the input key points to investigate performance drop. The proposed method is expected to show stable performance under such perturbation.
3. The proposed data completion strategy shows a marginal effect in Tab. 5.
4. Authors should discuss related work regarding efficient 3D human pose estimation [1, 2, 3] in Related Work.

[1] Zhao et al. PoseFormerV2: Exploring Frequency Domain for Efficient and Robust 3D Human Pose Estimation, 2023.

[2] Li et al. Exploiting Temporal Contexts with Strided Transformer for 3D Human Pose Estimation, 2022.

[3] Einfalt et al. Uplift and Upsample: Efficient 3D Human Pose Estimation with Uplifting Transformers, 2022.

**Questions:**

1. I'm confused about the statement in L78-80: "Furthermore, none of them cut to the biggest computational overhead – the Self Attention calculation which is quadratic to the number of tokens". The proposed method is similar to PoseFormerV2 [1] (and also other methods [2][3]) in terms of improving efficiency via a reduction in token number for self-attention (or frame number reduction).
2. How is the generalization ability of the proposed skipped transformer (how does it work if I replace the naive transformers in previous methods with skipped transformer)?

[1] Zhao et al. PoseFormerV2: Exploring Frequency Domain for Efficient and Robust 3D Human Pose Estimation, 2023.

[2] Li et al. Exploiting Temporal Contexts with Strided Transformer for 3D Human Pose Estimation, 2022.

[3] Einfalt et al. Uplift and Upsample: Efficient 3D Human Pose Estimation with Uplifting Transformers, 2022.

---

> ### Author Response · Authors · 2024-11-22
>
> **1. Quantitative Robustness Comparison**
>
> We have supplemented a detailed quantitative robustness comparison in Appendix **A.6 DETAILED QUANTITATIVE COMPARISON FOR ROBUSTNESS**. Please refer to Table 2 in the appendix for the experimental results on noise interference. The results demonstrate that G-SFormer consistently exhibits a more stable performance under varying noise intensities, further emphasizing its robustness against diverse types and levels of noisy input disturbances.
>
> **2. Efficacy of data completion methods**
>
> We have updated Table 5 to include the condition where neither SPE nor DR is employed. Please check the updated table for a more intuitive comparison.
>
> **3.  Distinctions of Skipped Self-Attention**
>
> G-SFormer and the listed methods all have an encoder-decoder framework, which might cause your confusion. Here, we clarify the distinctions:
>
> Encoding Stage:
>
> In G-SFormer, all temporal tokens are preserved, with longer contextual correlations established via Skipped Self-Attention (SSA). This approach significantly reduces computational cost by performing skipped attention calculations (refer to Equation 7 for details). In contrast, methods such as PoseFormerV2[1] and Uplift & Upsample[3] achieve computational cost reduction by decreasing the number of temporal tokens.
>
>
> Decoding Stage:
>
> While Strided Transformer[2] employs strided convolutional layers for temporal feature aggregation, G-SFormer directly conducts feature aggregation based on SSA, offering a more straightforward and efficient approach (refer to Equations 8-9 for details).
>
>
> **4. Efficacy of  Skipped Transformer**
>
> Due to the limited rebuttal time, we are currently unable to provide results for such experiments. However, in Table 6 of the Ablation Study, we evaluated the integration of Vanilla Transformer (VT) into G-SFormer as a replacement for Skipped Transformer. The VT-Conv and VT-Strided Conv show performance drop 1.0-1.3 mm with up to nearly twice computational cost, further verifying the superiority of the Skipped Transformer.

---

> > ### Comment · Reviewer_pC4k · 2024-11-24
> >
> > I appreciate the authors' efforts. However, I disagree with the claim that "G-SFormer consistently exhibits a more stable performance under varying noise intensities." When analyzing the performance drop (i.e., the difference in performance between conditions with and without noise), PoseFormerV2 appears to be more robust under slight noise levels (e.g., $\sigma = 0.1, 0.2$). While the proposed approach does show improved robustness under significant noise levels (e.g., $\sigma = 0.7, 0.8$), such scenarios may not be realistic. The experiment aims to simulate noise in 2D keypoints, but current state-of-the-art methods are unlikely to encounter such extreme noise levels.
> >
> > Given that my concerns regarding novelty, model design motivation, and contribution remain unresolved, I will maintain my current score.

---

> > > ### Author Response · Authors · 2024-11-25
> > >
> > > We sincerely thank the reviewer for the prompt response despite the busy schedule. We also fully understand the reviewer's concerns about the manuscript and their doubts regarding key issues. Here, we provide further clarification on these matters.
> > >
> > > **(1) Comparison with PoseFormerV2**
> > >
> > > We agree with the reviewer that PoseFormerV2 demonstrates robustness to low-amplitude noise in our experiments, primarily due to its use of low-frequency DCT in the temporal domain. However, this approach also filters out high-frequency temporal features, which contain valuable information beyond noise, leading to relatively lower overall performance. The relevant statements in the main manuscript are as follows:
> > >
> > > **line 076-082 in page 2**
> > > "Zhao et al. (2023) obtain low-frequency pose components through filtering out high-frequency noise and integrate them with sampled temporal features to generate a robust pose representation. ...... However, it is worth noting that reducing pose tokens in temporal domain will lead to performance degradation since partial information is inevitably lost... "
> > >
> > >
> > > **line 480-482 in page 12**
> > > "Although PoseFormerV2 (Zhao et al., 2023) extracts global-view representation by low-frequency DCT coefficients, this compression manner discards features in temporal domain and inevitably causes a loss of accuracy."
> > >
> > > In contrast, we propose the Skipped Transformer for temporal modeling, which avoids discarding temporal tokens to prevent accuracy loss. Instead, it employs Skipped Self-Attention (SSA) to enhance efficiency and capture long-range dynamics through multiple framesets. To further improve robustness against local joint deviations, we integrate the Part-based Adaptive GNN, which models global spatial correlations and generates a comprehensive pose feature for each frame. Overall, G-SFormer is designed to comprehensively achieve high performance, efficiency, and robustness through innovations in both spatial and temporal domains, setting it apart from PoseFormerV2 and other methods that reduce temporal tokens to improve efficiency or robustness.
> > >
> > > In addition to the Delta MPJPE metric, which indicates the degree of performance degradation, the overall MPJPE is also a crucial indicator of accuracy and robustness in 3D pose estimation. This is particularly important because **the detected 2D pose inputs inherently contain various types and levels of noise, as illustrated in Figure 1 of the main manuscript and Figure 1 of the Appendix.**
> > >
> > > G-SFormer demonstrates significantly higher accuracy compared to PoseFormerV2 across all large-to-small benchmarks. For instance, on the Human3.6M dataset, the low-cost version of G-SFormer-S achieves **1.9mm-2.5mm lower MPJPE** while using similar FLOPs and only 30.4%-35% of the parameters of PoseFormerV2. Moreover, The substantial superiority (lower MPJPE of 2.3-6mm) can also be revealed on 3DHP and Human-Eva datasets.
> > >
> > > However, as the reviewer reminds, we will change the statement of "In contrast, the proposed G-SFormer maintains more stable performance under varying noise intensities." in Appendix into :
> > > **"In contrast, the proposed G-SFormer shows more stable performance trend comprehensively considering both overall accuracy and performance degradation under varying noise intensities."**
> > > to make the statement more rigorous and consistent with experimental data.

---

> > > ### Author Response · Authors · 2024-11-25
> > >
> > > **(2) Motivation**
> > >
> > > We propose G-SFormer as a compact and adaptive framework to realize high accuracy, efficient and robust 3D pose reconstruction in a global approach. Unlike mainstream GNN-based and Transformer-based methods that rely heavily on dense joint-frame connections and iterative correlations, our design is based on larger granularities of body parts and framesets. This enables a global and comprehensive understanding of long-term dynamic motions and provides a streamlined solution for low-cost and robust 3D HPE.
> > >
> > > **(3) Novelty**
> > >
> > > The main novelty is based on our framework design:
> > >
> > > a. Part-based Adaptive GNN for spatial modeling:
> > >
> > > We propose a Part-based Adaptive GNN to model the coordination of human body and the interactions between body parts during movement. The generated graph feature is integrated with multi-granularity pose attributes to obtain a comprehensive pose representation for each frame. It is also a robust representation with less sensitivity to local joint deviations.
> > >
> > > b. Skipped Transformer for temporal modeling:
> > >
> > > The input pose sequence is sampled into framesets to implement Skipped Self-Attention (SSA) in parallel, establishing global-range alignments across multiple temporal perspectives. This design significantly enhances computational efficiency while capturing long-range dynamics of the lengthy input pose sequence.
> > >
> > > Additionally, we proposed two types of Data completion methods of SPE and DR to enhance the spatial structure and temporal motion information of input 2D poses, which are parameter-free and easy to implement.
> > >
> > > **(4) Contribution**
> > >
> > > **a.** Extensive experiments on Human3.6M, MPI-INF-3DHP and Human-Eva benchmarks demonstrate that G-SFormer series models can compete and outperform the state-of-the-arts with only a fraction of parameters, and exhibit outstanding efficiency in both the seq2seq and seq2frame workflows. G-SFormer provides an effective and practical solution for achieving steady and high-accuracy 3D HPE with a compact model size and minimal computational cost, making it particularly well-suited for applications on resource-limited platforms.
> > >
> > > **b.** We conducted an analysis of common errors in 2D pose detection and supplemented a quantitative evaluation of multi-level Gaussian noise disturbances to assess the robustness of G-SFormer. The stable performance trend further verifies its significant practical value for 3D HPE tasks in complex real-world scenarios.
> > >
> > > **c.** Trough the visualized attention map comparison, the proposed Skipped Self-Attention (SSA) exhibits sparse attention patterns distributed across a wider temporal range compared to the dense and concentrated self-attention in Vanilla Transformer. Furthermore, the supplementary comparison of SSA with other efficient attention mechanisms highlights its advantages in balancing accuracy and efficiency for sequence modeling. The above analysis indicates that the proposed SSA has potential for Transformer-based sequential modeling tasks.

---

> ### Comment · Reviewer_pC4k · 2024-11-25
>
> I would like to thank the authors for their efforts in response, but my concerns remain as follows:
> ### (1) Robustness
> Thank you for providing a detailed explanation of the comparison with PoseFormerV2. While the rationale makes sense, my primary concern regarding robustness lies in the mixed performance of the proposed method under weak and strong noise conditions.
>
> Firstly, I believe that relying only on absolute performance metrics is insufficient for evaluating robustness. Although the proposed model demonstrates strong baseline performance, the experiments in the supplementary material reveal that it is less robust to slight Gaussian noise compared to PoseFormerV2, as evidenced by a greater performance drop under these conditions.
>
> Conversely, I acknowledge that the proposed method exhibits relatively stable performance when handling missing keypoints (corresponding to Gaussian noise with large magnitude), as also demonstrated by the qualitative results. This suggests that the method can handle certain extreme scenarios effectively.
>
> However, these mixed results do not provide a convincing case for the robustness of the proposed method. The underlying factors contributing to the observed robustness—or its limitations—remain unclear.
>
> ### (2) Novelty
> After considering the response from Reviewer U2V7, I still find the novelty of this work to be incremental.
>
> ### (3) Contributions
> Thank you for outlining the contributions of this work. Upon review, I have some concerns regarding the distinctiveness and clarity of the claimed contributions:
>
> Claim (b): The evaluation of multi-level Gaussian noise has already been addressed in prior work [1, 2]. Furthermore, the robustness of the proposed method, as discussed earlier (see (1) Robustness), appears to be weaker compared to existing methods. This raises questions about the novelty and significance of this contribution in the context of prior research.
>
> Taking into account the strengths of the empirical results alongside the concerns mentioned above, I have decided to raise my score to 5. However, I do not feel that a higher score is justified at this time, given the remaining issues related to the novelty and impact of the work.

---

> > ### Author Response · Authors · 2024-12-01
> >
> > We sincerely appreciate the reviewer’s patient and thorough feedback, which has encouraged us to re-examine our work with greater precision and rigor.
> >
> > **1.$\Delta$MPJPE compared with PoseFormerV2**
> >
> > To analyze the focusing issue of the robustness of G-SFormer, particularly the $\Delta$MPJPE compared to PoseFormerV2, we re-edited Table 2 in the Appendix, explicitly listing $\Delta$MPJPE to provide a clearer comparison.
> >
> > It can be seen that although the overall accuracy of G-SFormer maintains higher compared with PoseFormerV2, its performance degradation is larger when $\Delta$MPJPE is within the lower range of 0.1-0.2. However, the situation reverses when the noise deviation $\delta$ exceeds 0.4, at which point PoseFormerV2 shows severe performance degradation as the noise intensity grows.
> >
> > | $\delta$ | G-SFormer | $\Delta$ MPJPE | G-SFormer-L | $\Delta$ MPJPE | PoseFormerV2 | $\Delta$ MPJPE | MixSTE | STCFormer |
> > |----------|-----------|----------------|-------------|----------------|--------------|----------------|--------|-----------|
> > | 0        | 42.3      | -              | 41.6        | -              | 45.2         | -              | 40.9   | 42.0      |
> > | 0.1      | 44.8      | 2.5            | 44.2        | 2.6            | 46.3         | 1.1            | 54.6   | 47.5      |
> > | 0.2      | 48.8      | 6.5            | 49.4        | 7.8            | 48.9         | 3.7            | 66.3   | 55.2      |
> > | 0.4      | 55.9      | 13.6           | 54.4        | 12.8           | 59.2         | 14.0             | 78.5   | 70.6      |
> > | 0.5      | 58.5      | 16.2           | 62.7        | 21.1           | 66.6         | 21.4           | 82.2   | 77.9      |
> > | 0.7      | 63.5      | 21.2           | 69.3        | 27.7           | 83.5         | 38.3           | 86.9   | 91.1      |
> > | 0.8      | 65.6      | 23.3           | 72.2        | 30.6           | 91.7         | 46.5           | 88.5   | 97.0      |
> > | 1.0      | 68.4      | 26.1           | 77.0        | 35.4           | 105.8        | 60.6           | 90.5   | 107.5     |
> >
> > However, **could the noise strength be excessively large, making G-SFormer only applicable to rare cases?** After further investigation and analysis, we provide the following clarification:
> >
> > Referring to the noise disturbance experiments in PoseFormerV2 (e.g., Figures 6, 9, and 10), the standard deviation ($\delta$) range of the noise is set within 1–10. In contrast, our range is 0–1. **This confirms that our evaluated noise strength is within a reasonable range, rather than being unrealistically high.**
> >
> > But it can also be observed that the maximum $\Delta$MPJPE presented in the figures of PoseformerV2 is much smaller comparted to our experimental results. This discrepancy may arise from differences in experimental settings. As detailed in the appendix, we randomly choose 24 frames with 2 joints in the input pose sequece to add Gaussian noise. **Here we reduce the number of noisy frames to 12 and conduct noise interference experiment again**. Although the overall MPJPE and $\Delta$MPJPE are significantly reduced, the same variation trend can still be observed.
> >
> > | $\delta$ | G-SFormer | $\Delta$ MPJPE | PoseFormerV2 | $\Delta$ MPJPE |
> > |----------|-----------|----------------|--------------|----------------|
> > | 0        | 42.3      | -              | 45.2         | -              |
> > | 0.1      | 43.4      | 1.1           |  45.7         | 0.5           |
> > | 0.2      | 45.1     |2.8            | 46.9         | 1.7           |
> > | 0.4      | 48.5      | 6.2           | 51.7         | 6.5             |
> > | 0.5      | 49.7     | 7.4      |  55.5         | 10.3         |
> > | 0.7      | 51.8     |9.5           |  65.2        | 20.0          |
> > | 0.8      | 52.9    | 10.6           |  70.8         |25.6           |
> > | 1.0      | 54.7      | 12.4           | 82.4       | 37.2           |
> >
> > Therefore, when focusing solely on the $\Delta$MPJPE metric, the proposed method still demonstrates an advantage in robustness, as the open interval of standard deviation beyond 0.4 covers a much wider range of noise in real-world scenarios. Additionally, PoseFormerV2 shows significantly more severe performance degradation compared to its earlier advantage. **This trend highlights the robustness of the proposed method, making it more reliable in real-world applications where noise intensities can vary widely.**

---

> > ### Author Response · Authors · 2024-12-01
> >
> > **2.	Robustness in Qualitative Study**
> >
> > For the qualitative study, we agree with the reviewer’s observation that “the proposed method exhibits relatively stable performance when handling missing keypoints ”. Real-world noise brought by detection errors is indeed complex and includes diverse types, as described in line 473-475:
> >
> > “*Typical hard cases are presented in Figure 1(b), including detection errors such as joint position deviation, left-right switch, confusion caused by self-occlusion and miss/coincidence detection. G-SFormer estimates more refined and structurally reasonable 3D poses than the competitors.*”
> >
> > Based on the qualitative analysis with related cases presented in Figure 1 of the manuscript and Figure 1 of the Appendix, G-SFormer outperforms PoseFormerV2 in handling challenging cases with missing or erroneous joints .
> >
> > **3.	The claim in Robustness**
> >
> > Based on the above results, G-SFormer demonstrates stable performance and robustness across diverse types and levels of noise. **This advantage cannot be replaced by PoseFormerV2, which is limited to a narrow range of noise interference and deteriorates sharply under widespread noise intensities.** This aligns with the claims made in the manuscript regarding G-SFormer’s robustness:
> >
> > **Line 031-032**:
> > “*It also exhibits outstanding robustness to inaccurately detected 2D poses.*”
> >
> > **Line 482-483**:
> > “*G-SFormer exhibits stable performance in terms of both accuracy and robustness against missing or erroneous joints.*”
> >
> > The quantitative analysis further confirms the robustness claim of G-SFormer. We will include a comprehensive presentation of the quantitative comparison results in the main manuscript, to enhance the objectivity of our robustness discussion.
> >
> > **Line484 in section 4.3.2:**
> >
> > *From the robustness quantitative study, it is worth noting that PoseFormerV2 shows less performance degradation when the noise strength is within a small-range (e.g., $\delta$=0.1, 0.2), but  deteriorates sharply as the noise increases. In contrast, G-SFormer maintains a more stable performance trend considering both overall accuracy and performance variation across widespread noise intensities, making it more reliable in dealing with complex noise in real-world applications.*

---

### Meta-Review · Area_Chair_8Hab · 2024-12-16

**Metareview:**

This paper introduces G-SFormer, a new method combining a Part-based Adaptive Graph Neural Network (GNN) and a Frame set-based Skipped Transformer for efficient and robust 2D-to-3D human pose estimation. It achieves competitive performance on multiple benchmarks while reducing computational cost through adaptive spatial modelling and frame-skipping strategies. The reviewers highlighted critical issues with the paper, including concerns that the proposed combination of a part-based GNN and a skipped transformer appears incremental and lacks clear motivation for its robustness against noisy 2D key-points. The paper's structure was deemed disorganised, making the core contributions difficult to follow, and the claimed >100-fold efficiency speedup was not adequately explained with detailed analysis. Additionally, the seq2frame pipeline was criticised for its limited applicability in practical scenarios.

**Additional Comments On Reviewer Discussion:**

Reviewers identified critical issues with the paper. Reviewers pC4k, kpzv, and Xnmp noted that the proposed combination of a part-based GNN and a skipped transformer appears incremental and lacks a clear justification for its robustness against noisy 2D key points (highlighted by Reviewer pC4k). The paper's structure was described as disorganised, making it difficult to discern the core contributions, and the claimed efficiency improvement was not adequately supported with detailed analysis (as pointed out by Reviewer kpzv). Furthermore, Reviewer Xnmp criticised the seq2frame pipeline for its limited applicability in practical scenarios. Although the rebuttal addressed several other concerns raised by the reviewers, significant issues remain unresolved, including limited novelty, the unclear motivation for robustness to noisy 2D key points, and the lack of explanation for the reported 138-fold speedup over MixSTE.
In summary, after the rebuttal, one reviewer recommended "Reject," and four reviewers rated the paper as "marginally below the acceptance threshold." Reviewer Xnmp rated it "marginally above the acceptance threshold" but did not strongly advocate for acceptance, citing concerns about the limited practical applicability of the seq2frame pipeline, the limited novelty of the proposed method, and shared concerns with Reviewer pC4k regarding its robustness.

---

### Decision · Program_Chairs · 2025-01-22

Reject